# FedL2G: Learning to Guide Local Training in Heterogeneous Federated Learning

## Abstract

Data and model heterogeneity are two core issues in Heterogeneous Federated Learning (HtFL). In scenarios with heterogeneous model architectures, aggregating model parameters becomes infeasible, leading to the use of prototypes (*i.e.*, class representative feature vectors) for aggregation and guidance. However, when aligned with global prototypes, they still experience a mismatch between the extra guiding objective and the client's original local objective. Thus, we propose a Federated Learning-to-Guide (FedL2G[1]) method that adaptively learns to guide local training in a federated manner and ensures the extra guidance is beneficial to clients' original tasks. With theoretical guarantees, FedL2G efficiently implements the learning-to-guide process using only first-order derivatives *w.r.t.* model parameters and achieves a non-convex convergence rate of $\mathcal{O}(1/T)$. We conduct extensive experiments on two data heterogeneity and six model heterogeneity settings using 14 heterogeneous model architectures (*e.g.*, CNNs and ViTs) to demonstrate FedL2G's superior performance compared to six counterparts.

## 1 Introduction

With the rapid development of AI techniques (Touvron et al., 2023; Achiam et al., 2023), public data has been consumed gradually, raising the need to access local data inside devices or institutions (Ye et al., 2024). However, directly using local data often raises privacy concerns (Nguyen et al., 2021). Federated Learning (FL) is a promising privacy-preserving approach that enables collaborative model training across multiple clients (devices or institutions) in a distributed manner without the need to move the actual data outside clients (Kairouz et al., 2019; Li et al., 2020). Nevertheless, data heterogeneity (Li et al., 2021; Zhang et al., 2023d;a) and model heterogeneity (Zhang et al., 2024b; Yi et al., 2023) remain two practical issues when deploying FL systems. Personalized FL (PFL) mainly focuses on the data heterogeneity issue (Zhang et al., 2023e), while Heterogeneous FL (HtFL) considers both data and model heterogeneity simultaneously (Zhang et al., 2024a). HtFL's support for model heterogeneity enables a broader range of clients to participate in FL with their customized models.

In HtFL, sharing model parameters, a widely used technique in traditional FL and PFL is not applicable (Zhang et al., 2024b). Instead, lightweight knowledge carriers, including small auxiliary models (Shen et al., 2020; Wu et al., 2022; Yi et al., 2024), tiny homogeneous modules (Liang et al., 2020; Yi et al., 2023), and prototypes (*i.e.*, class representative feature vectors) (Jeong et al., 2018; Tan et al., 2022b), can be shared among clients. Prototypes offer the most significant communication efficiency due to their compact size.

However, representative prototype-based methods FedDistill (Jeong et al., 2018) and FedProto (Tan et al., 2022b), still suffer from a mismatch between the prototype-guiding objective and the client's original local objective. These methods typically introduce an extra guiding objective alongside the original local objective, aiming to guide local features to align with the global ensemble prototypes. Due to the significant variation in width and depth among clients' heterogeneous models, their feature extraction capabilities also differ considerably (Zhang et al., 2024a;b). On the other hand, the data distribution also diverges across clients (McMahan et al., 2017; Li et al., 2022). Since the global prototypes are derived from aggregating diverse local prototypes, they inherently cannot fully align

---

[1]Code is included in the supplementary material.

with specific client models and their respective data. Consequently, directly optimizing the guiding and local objectives together *without* prioritizing the original local objective has the potential to undermine the local objective of each client due to the objective mismatch, as shown in Fig. 1.

To address the issue of objective mismatch, we propose a novel **Federated Learning-to-Guide (`FedL2G`)** method. It prioritizes the original local objective while learning the guiding objective, ensuring that the guiding objective facilitates each client's original local task rather than causing negative effects to the original local objective. This is why we term it "*learning to guide*". Specifically, we hold out a tiny *quiz set* from the training set and denote the remaining set as a *study set* on each client. Then we learn guiding vectors in a federated manner, ensuring that updating client models with the extra guiding loss and the original local loss on their study sets consistently reduces the original local loss on their quiz sets (which are not used for training and testing). The steadily decreasing original local loss (no loss increase) and the superior test accuracy illustrated in Fig. 1 embody the design philosophy and effectiveness of our `FedL2G`. Moreover, in contrast to learning-to-learn (Finn et al., 2017; Jiang et al., 2019; Fallah et al., 2020a), the learning-to-guide process in our `FedL2G` only requires first-order derivatives *w.r.t.* model parameters, making it computationally efficient.

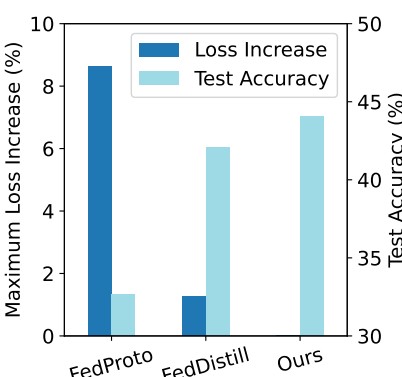

Figure 1: The objective mismatch problem increases the original local loss during FL, leading to lower test accuracy. The *loss increase* is calculated as the difference between the current original local loss and its previous minimum.

We assess the performance of our `FedL2G` across various scenarios, including two types of data heterogeneity, six types of model heterogeneity involving 14 different model architectures, and diverse system settings. In addition to test accuracy, we also evaluate communication and computation overhead. The results consistently demonstrate that `FedL2G` outperforms six state-of-the-art methods, achieving the best model performance. We summarize our contributions as follows:

- In the context of HtFL with data and model heterogeneity, we analyze and observe the objective mismatch issue between the extra guiding objective and the original local objective within representative prototype-based methods.

- We propose a `FedL2G` method that prioritizes the original local objective while using the extra guiding objective to eliminate the objective mismatch issue.

- We theoretically prove that `FedL2G` achieves efficiency using only first-order derivatives *w.r.t.* model parameters, with a non-convex convergence rate of $\mathcal{O}(1/T)$.

- To demonstrate our `FedL2G` 's priority, we conducted extensive experiments covering two types of data heterogeneity, six types of model heterogeneity (including 14 distinct model architectures such as CNNs and ViTs), and various system settings.

## 2 RELATED WORK

### 2.1 HETEROGENEOUS FEDERATED LEARNING (HTFL)

Presently, FL is one of the popular collaborative learning and privacy-preserving techniques (Zhang et al., 2023d; Li et al., 2020) and HtFL extends traditional FL by supporting model heterogeneity (Ye et al., 2023). Prevailing HtFL methods primarily consider three types of model heterogeneity: (1) group heterogeneity, (2) partial heterogeneity, and (3) full heterogeneity (Zhang et al., 2024b). Among them, the HtFL methods considering group model heterogeneity extract different but architecture-constraint sub-models from a global model for various groups of clients (Diao et al., 2020; Horvath et al., 2021; Wen et al., 2022; Luo et al., 2023; Zhou et al., 2023). Thus, they cannot support customized client models and are *excluded* from our consideration. Additionally, sharing and revealing model architectures within each group of clients also raises privacy and intellectual property concerns (Zhang et al., 2024a). As the server is mainly utilized for parameter aggregation in prior FL systems (Tan et al., 2022a; Kairouz et al., 2019), training a server module with a large

number of epochs, like (Zhang et al., 2024b;a; Zhu et al., 2021), necessitates additional upgrades or the purchase of a new heavy server, which is impractical. Thus, we focus on the *server-lightweight* methods.

Both partial and full model heterogeneity accommodate customized client model architectures, but partial heterogeneity still assumes that some small parts of all client models are homogeneous. For example, LG-FedAvg (Liang et al., 2020) and FedGH (Yi et al., 2023) stand out as two representative approaches. LG-FedAvg and FedGH partition each client model into a feature extractor part and a classifier head part, operating under the assumption that all classifier heads are homogeneous. In LG-FedAvg, the parameters of classifier heads are uploaded to the server for aggregation. In contrast, FedGH uploads prototypes to the server and trains the lightweight global classifier head for a small number of epochs. Both methods utilize the global head for knowledge transfer among clients but overlook the inconsistency between the global head and local tasks.

In the case of full model heterogeneity, mutual distillation (Zhang et al., 2018) and prototype guidance (Tan et al., 2022b) emerge as two prevalent techniques. Using mutual distillation, FML (Shen et al., 2020) and FedKD (Wu et al., 2022) facilitate client knowledge transfer through a globally shared auxiliary model. However, sharing an entire model demands substantial communication resources, even if the auxiliary model is typically small (Zhang et al., 2024b). Furthermore, aggregating a global model in scenarios with data heterogeneity presents numerous challenges, such as client-drift (Karimireddy et al., 2020), ultimately leading to a subpar global model (Li et al., 2022; Zhang et al., 2023a;b;c). As representative prototype guidance methods, FedDistill (Jeong et al., 2018) and FedProto (Tan et al., 2022b) gather prototypes on each client, aggregate them on the server to create the global prototypes, and guide client local training with these global prototypes. Specifically, FedDistill extracts lower-dimensional prototypes than FedProto. This difference stems from FedDistill applying prototype guidance in the logit space, whereas FedProto uses the intermediate feature space. Sharing higher-dimensional prototypes can transfer more information among clients but may also exacerbate the negative effects of objective mismatch.

## 2.2 STUDENT-CENTERED GUIDANCE

Our learning-to-guide philosophy draws inspiration from student-centered knowledge distillation approaches (Yang et al., 2024). They are based on the insight that a teacher's subject matter expertise alone may not match the student's specific studying ability and style, resulting in negative effects (Sengupta et al., 2023; Yang et al., 2024). To address the mismatch between the teacher's knowledge and the needs of the student, updating the teacher model with concise feedback from the student on a small quiz set represents a promising direction (Ma et al., 2022; Zhou et al., 2022; Sengupta et al., 2023).

However, these student-centered approaches are built upon a teacher-student framework, assuming the presence of a well-trained large teacher model. They concentrate on a central training scheme without factoring in distributed multiple students and privacy protection (Lee et al., 2022; Hu et al., 2022), rendering them inapplicable in the context of HtFL. Additionally, modifying and extending these student-centered approaches to HtFL requires significant communication and computational resources to update a shared large teacher model based on student feedback (Zhou et al., 2022; Lu et al., 2023). Nevertheless, the student-centered guidance concept inspires us to propose a learning-to-guide approach in HtFL. This involves substituting the large teacher model with compact guiding vectors and updating these guiding vectors based on clients' feedback from their quiz sets, making our `FedL2G` lightweight, efficient, and adaptable.

## 3 FEDERATED LEARNING-TO-GUIDE: FEDL2G

### 3.1 NOTATIONS AND PRELIMINARIES

**Problem statement.** In an HtFL system, $N$ clients, on the one hand, train their heterogeneous local models (with parameters $\theta_1, \ldots, \theta_N$) using their private and heterogeneous training data $\mathcal{D}_1, \ldots, \mathcal{D}_N$. On the other hand, they share some global information, denoted by $\mathcal{G}$, with the

assistance of a server to facilitate collaborative learning. Formally, the typical objective of HtFL is

$$\min_{\boldsymbol{\theta}_1,\ldots,\boldsymbol{\theta}_N} \sum_{i=1}^{N} \frac{|\mathcal{D}_i|}{D} \mathcal{L}_{\mathcal{D}_i}(\boldsymbol{\theta}_i, \mathcal{G}), \tag{1}$$

where $|\mathcal{D}_i|$ represents the size of the training set $\mathcal{D}_i$, $D = \sum_{i=1}^{N} |\mathcal{D}_i|$, and $\mathcal{L}_{\mathcal{D}_i}$ denotes a total client training objective over $\mathcal{D}_i$.

**Prototype-based HtFL.** Sharing class-wise prototypes of low-dimensional features in either the intermediate feature space or the logit space among clients has become a prevalent and communication-efficient solution to address model heterogeneity in HtFL (Ye et al., 2023). Take the popular scheme (Jeong et al., 2018) for example, where prototypes are shared in the logit space, $\mathcal{G}$ (the set of global prototypes) is defined by

$$\mathcal{G} = \{\boldsymbol{g}^y\}_{y=1}^{C}, \quad \boldsymbol{g}^y = agg(\{\boldsymbol{g}_1^y, \ldots, \boldsymbol{g}_N^y\}), \quad \boldsymbol{g}_i^y = \mathbb{E}_{(\boldsymbol{x},y)\sim\mathcal{D}_{i,y}}[f_i(\boldsymbol{x}, \boldsymbol{\theta}_i)], \tag{2}$$

and $C$ represents the total number of clients' original local tasks classes. $\boldsymbol{g}^y$ and $\boldsymbol{g}_i^y$ denote the global and local prototypes of class $y$, respectively. Besides, $agg$ is an aggregation function defined by each prototype-based HtFL method, $\mathcal{D}_{i,y}$ stands for a subset of $\mathcal{D}_i$ containing all the data of class $y$, and $f_i$ represents the local model of client $i$. Given a global $\mathcal{G}$, client $i$ then takes prototype guidance for knowledge transfer among clients via

$$\mathcal{L}_{\mathcal{D}_i}(\boldsymbol{\theta}_i, \mathcal{G}) := \mathbb{E}_{(\boldsymbol{x},y)\sim\mathcal{D}_i}[\ell_{ce}(f_i(\boldsymbol{x}, \boldsymbol{\theta}_i), y) + \ell_g(f_i(\boldsymbol{x}, \boldsymbol{\theta}_i), \boldsymbol{g}^y)], \tag{3}$$

where the weight of $\ell_g$ is set to one to balance two objectives equally here, $\ell_{ce}$ is the original local cross-entropy loss (Zhang & Sabuncu, 2018), and $\ell_g$ is the guiding loss.

## 3.2 LEARNING TO GUIDE

**Motivation.** Initially, heterogeneous client models trained by $\ell_{ce}$ can adapt to their local data with diverse feature extraction capabilities. However, directly adding $\ell_g$ *without* prioritizing $\ell_{ce}$ can cause the model of each client to deviate from $\ell_{ce}$. On the other hand, since all feature vectors are extracted on heterogeneous client data, the aggregated global prototype, *e.g.*, $\boldsymbol{g}^y$, is data-derived, which may deviate from the features regarding class $y$ on each client. Both the model and data heterogeneity result in the objective mismatch issue between $\ell_{ce}$ and $\ell_g$, which causes the negative effect to $\ell_{ce}$ when using $\ell_g$, as shown in Fig. 1 and discussed further in Sec. 4.5. Therefore, we propose a novel FedL2G method, which substitutes the data-derived prototypes with trainable guiding vectors $\mathcal{G} = \{\boldsymbol{v}^y\}_{y=1}^{C}$ and ensures that $\mathcal{G}$ *is learned to reduce* $\ell_{ce}$ *when guided by* $\ell_g$. Formally, we replace Eq. (3) with a new loss to train the client model:

$$\mathcal{L}_{\mathcal{D}_i}(\boldsymbol{\theta}_i, \mathcal{G}) := \mathbb{E}_{(\boldsymbol{x},y)\sim\mathcal{D}_i}[\ell_{ce}(f_i(\boldsymbol{x}, \boldsymbol{\theta}_i), y) + \ell_g(f_i(\boldsymbol{x}, \boldsymbol{\theta}_i), \boldsymbol{v}^y)], \tag{4}$$

where the learning of guiding vectors $\mathcal{G}$ is the key step.

**Learning guiding vectors.** Without relying on data-derived information, we randomly initialize the global $\mathcal{G}$ on the server and update it based on the aggregated gradients from participating clients in each communication iteration. Inspired by the technique of outer-inner loops in meta-learning (Zhou et al., 2022), we derive the gradients of client-specific $\boldsymbol{v}_i^y$ in the *outer-loop*, while focusing on reducing the original local loss, *i.e.*, $\ell_{ce}$, in the *inner-loop* on each client. To implement the learning-to-guide process, we hold out a tiny *quiz set* $\mathcal{D}_i^q$ (one batch of data) from $\mathcal{D}_i$ and denote the remaining training set as the *study set* $\mathcal{D}_i^s$. Notice that we exclusively conduct model updates on $\mathcal{D}_i^s$ and never train $\boldsymbol{\theta}_i$ on $\mathcal{D}_i^q$. In particular, $\mathcal{D}_i^q$ is solely used to evaluate $\boldsymbol{\theta}_i$'s performance regarding the original local loss and derive the gradients (feedback) *w.r.t.* $\boldsymbol{v}_i^y$. Below, we describe the details of FedL2G in the $t$-th iteration, using the notation $t$ solely for the global $\mathcal{G}$ for clarity. Recall that $\mathcal{G} = \{\boldsymbol{v}^y\}_{y=1}^{C}$, we use the general notation $\mathcal{G}$ in the following descriptions for simplicity, although all operations correspond to each $\boldsymbol{v}^y, y \in \{1, \ldots, C\}$ within $\mathcal{G}$.

Firstly, in step ①, we download $\mathcal{G}^{t-1}$ from the server to client $i$. Then, in step ②, we perform regular training for $\boldsymbol{\theta}_i$ on $\mathcal{D}_i^s$ using $\mathcal{L}_{\mathcal{D}_i^s}(\boldsymbol{\theta}_i, \mathcal{G}^{t-1})$ (see Eq. (4)). Sequentially, the pivotal steps ③ and ④ correspond to our objective of learning-to-guide. We illustrate steps ③ and ④ in Fig. 2. In step ③, we execute a *pseudo-train* step (without saving the updated model back to disk) on a randomly sampled batch $\mathcal{B}_i^s$ from $\mathcal{D}_i^s$, *i.e.*,

$$\boldsymbol{\theta}_i'(\mathcal{G}^{t-1}) \leftarrow \boldsymbol{\theta}_i - \eta_c \nabla_{\boldsymbol{\theta}_i} \mathcal{L}_{\mathcal{B}_i^s}(\boldsymbol{\theta}_i, \mathcal{G}^{t-1}), \tag{5}$$

where $\eta_c$ is the client learning rate, and we call $\boldsymbol{\theta}'_i(\mathcal{G}^{t-1})$ as the pseudo-trained local model parameters, which is a function of $\mathcal{G}^{t-1}$. In step ④, our aim is to update the $\mathcal{G}^{t-1}$ in $\mathcal{L}_{\mathcal{B}^s_i}(\boldsymbol{\theta}_i, \mathcal{G}^{t-1})$ (see Eq. (4)) to minimize $\ell_{ce}$ with $\boldsymbol{\theta}'_i(\mathcal{G}^{t-1})$ on $\mathcal{D}^q_i$, thus we compute the gradients of $\mathcal{G}^{t-1}$ w.r.t. $\ell_{ce}$ on $\mathcal{D}^q_i$: $\nabla_{\mathcal{G}^{t-1}}\mathbb{E}_{(\boldsymbol{x},y)\sim\mathcal{D}^q_i}[\ell_{ce}(f_i(\boldsymbol{x}, \boldsymbol{\theta}'_i(\mathcal{G}^{t-1})), y)]$ (see Sec. 3.3 for details). Afterwards, we upload clients' gradients of $\mathcal{G}^{t-1}$ in step ⑤ and aggregate them in step ⑥. Then, in step ⑦, we update the global $\mathcal{G}^{t-1}$ on the server with the aggregated gradients. Put steps ③, ④, ⑤, ⑥, ⑦ together, we have

$$\mathcal{G}^t = \mathcal{G}^{t-1} - \eta_s \frac{1}{|\mathcal{I}^t|} \sum_{i\in\mathcal{I}^t} \nabla_{\mathcal{G}^{t-1}}\mathbb{E}_{(\boldsymbol{x},y)\sim\mathcal{D}^q_i}[\ell_{ce}(f_i(\boldsymbol{x}, \boldsymbol{\theta}_i - \eta_c\nabla_{\boldsymbol{\theta}_i}\mathcal{L}_{\mathcal{B}^s_i}(\boldsymbol{\theta}_i, \mathcal{G}^{t-1})), y)], \quad (6)$$

where $\eta_s$ is the server learning rate and $\mathcal{I}^t$ is the set of participating clients in the $t$-th iteration. We utilize the weight $\frac{1}{|\mathcal{I}^t|}$ here, considering that all participating clients execute step ③ and ④ with identical sizes of $\mathcal{B}^s_i$ and $\mathcal{D}^q_i$, $i \in \{1, \ldots, N\}$. Since some classes may be absent on certain clients, we only upload and aggregate the non-zero gradient vectors to minimize communication costs. We can easily implement Eq. (6) using popular public tools, *e.g.*, higher (Grefenstette et al., 2019).

**Warm-up period.** Since $\mathcal{G}$ is randomly initialized, using an uninformative $\mathcal{G}$ misguides local model training in Eq. (4). Therefore, before conducting regular client training in step ②, FedL2G requires a warm-up period of $T'$ iterations with step ①, ③, ④, ⑤, ⑥, ⑦. Without step ②, the warm-up process only involves one batch of each client's quiz set, thus demanding relatively small computation overhead.

**Twin HtFL methods based on FedL2G.** The above processes assume sharing information in the logit space, denoted as FedL2G-l. Additionally, when considering the intermediate feature space, we can rephrase all the corresponding $\ell_g$, for instance, rewriting $\ell_g(h_i(\boldsymbol{x}, \boldsymbol{\theta}^h_i), \boldsymbol{v}^y)$ in Eq. (4), where $h_i$ represents the feature extractor component in $f_i$, $\boldsymbol{\theta}^h_i \subset \boldsymbol{\theta}_i$ denotes the associated model parameters, and $\boldsymbol{v}^y$ resides in the intermediate feature space. We denote this twin method as FedL2G-f. The server learning rate $\eta_s$ is the *unique* hyperparameter in our FedL2G-l or FedL2G-f. Due to space constraints, we offer a detailed algorithm of our FedL2G-l in Algorithm 1. Extending it to FedL2G-f only requires necessary substitutions.

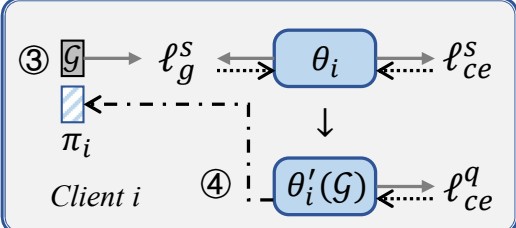

Figure 2: Steps ③ and ④ of FedL2G in one iteration. $\ell^s$ and $\ell^q$ denote the loss computed on the study and quiz sets, respectively. We omit data for clarity. Step ③: Update the local model $\boldsymbol{\theta}_i$ to a *pseudo model* $\boldsymbol{\theta}'_i(\mathcal{G})$ (parameterized by $\mathcal{G}$) using the study set. Step ④: Compute the gradients of $\mathcal{G}$ from Step ③ based on the gradients of $\boldsymbol{\theta}'_i(\mathcal{G})$ to minimize $\ell^q_{ce}$ on the quiz set.

### 3.3 EFFICIENCY ANALYSIS

As we compute gradients for two different entities in the *outer-loop* and *inner-loop*, respectively, we eliminate the necessity for calculating the second-order gradients of model parameters *w.r.t.* $\ell_{ce}$ as well as the associated computationally intensive Hessian (Fallah et al., 2020b). Our analysis is founded on Assumption 1 and Assumption 2 in Appendix C. Due to space limit, we leave the derivative details to Eq. (C.11) and show client $i$'s gradient *w.r.t.* $\mathcal{G}$ here:

$$\pi_i = -\eta_c\mathbb{E}_{(\boldsymbol{x},y)\sim\mathcal{D}^q_i}\{\nabla_1\ell_{ce} \cdot \nabla_2 f_i \cdot \mathbb{E}_{(\boldsymbol{x}',y')\sim\mathcal{B}^s_i}[\nabla_2 f_i \cdot \nabla_{\mathcal{G}^{t-1}}\nabla_1\ell_g]\}, \quad (7)$$

where $\nabla_1\ell_{ce} := \nabla_{a_1}\ell_{ce}(a_1, a_2)$, indicating the derivative of $\ell_{ce}$ *w.r.t.* the first variable, and so for $\nabla_2 f_i$ and $\nabla_1\ell_g$. The operation $\cdot$ denotes multiplication. Computing $\nabla_1\ell_{ce}$ and $\nabla_2 f_i$ is a common practice in deep learning (Zhang & Sabuncu, 2018) and calculating the $\nabla_{\mathcal{G}^{t-1}}\nabla_1\ell_g$ term is pivotal. To simplify the calculation, we choose the MSE loss as our $\ell_g$, so $\ell_g(f_i(\boldsymbol{x}', \boldsymbol{\theta}_i), \boldsymbol{v}^{y'}) = \frac{1}{M}\sum_{m=1}^M[f_i(\boldsymbol{x}', \boldsymbol{\theta}_i)_m - \boldsymbol{v}^{y'}_m]^2$, where $M$ is the dimension of $\boldsymbol{v}^{y'}$. Given $\mathcal{G} = \{\boldsymbol{v}^y\}^C_{y=1}$, we have

$$\nabla_{\mathcal{G}^{t-1}}\nabla_1\ell_g = \frac{2}{M}\sum_{m=1}^M \nabla_{\mathcal{G}^{t-1}}(f_i(\boldsymbol{x}', \boldsymbol{\theta}_i)_m - \boldsymbol{v}^{y'}_m) = \frac{2}{M}\sum_{m=1}^M -1 = -2. \quad (8)$$

---

**Algorithm 1** The Learning Processes in `FedL2G-l`

---

**Input:** $N$ clients; initial parameters $\boldsymbol{\theta}_1^0, \ldots, \boldsymbol{\theta}_N^0$ and $\mathcal{G}^0 = \{\boldsymbol{v}^{y,0}\}_{y=1}^C$; $\eta_c$: local learning rate; $\eta_s$: server learning rate; $\rho$: client joining ratio; $E$: local epochs; $T$: total communication iterations.

**Output:** Well-trained client model parameters $\boldsymbol{\theta}_1, \ldots, \boldsymbol{\theta}_N$.

1: All clients split their training data into a study set $\mathcal{D}^s$ and a batch of quiz set $\mathcal{D}^q$.
2: **for** communication iteration $t = 1, \ldots, T$ **do**
3:      Server samples a client subset $\mathcal{I}^t$ based on $\rho$.
4:      Server sends $\mathcal{G}^{t-1}$ to each client in $\mathcal{I}^t$.
5:      **for** Client $i \in \mathcal{I}^t$ in parallel **do**
6:          **if** $t > T'$ **then**
7:              Updates $\boldsymbol{\theta}_i^{t-1}$ to $\boldsymbol{\theta}_i^t$ using SGD for $E$ epochs via
                 $\min_{\boldsymbol{\theta}_i} \ \mathbb{E}_{(\boldsymbol{x},y)\sim\mathcal{D}_i^s}[\ell_{ce}(f_i(\boldsymbol{x}, \boldsymbol{\theta}_i^{t-1}), y) + \ell_g(f_i(\boldsymbol{x}, \boldsymbol{\theta}_i^{t-1}), \boldsymbol{v}^{y,t-1})]$
8:          **else**
9:              Marks $\boldsymbol{\theta}_i^{t-1}$ as $\boldsymbol{\theta}_i^t$.
10:          Executes a *pseudo-train* step on a randomly sampled batch $\mathcal{B}_i^s$ via Eq. (5) with $\boldsymbol{\theta}_i^t$.
11:          Computes the gradients of $\mathcal{G}^{t-1}$, *i.e.*, $\pi_i^t$, on $\mathcal{D}_i^q$ via Eq. (7).
12:          Sends non-zero vectors among $\pi_i^t$ to the server.
13:      Server averages the non-zero vectors of $\pi_i^t, i \in \mathcal{I}^t$ for each class to obtain $\pi^t$.
14:      Server updates $\mathcal{G}^{t-1}$ to $\mathcal{G}^t$ via $\mathcal{G}^t = \mathcal{G}^{t-1} - \eta_s \pi^t$.
15: **return** $\boldsymbol{\theta}_1^T, \ldots, \boldsymbol{\theta}_N^T$.

---

Finally, we obtain

$$\pi_i = 2\eta_c \mathbb{E}_{(\boldsymbol{x},y)\sim\mathcal{D}_i^q}\{\nabla_1\ell_{ce} \cdot \nabla_2 f_i \cdot \mathbb{E}_{(\boldsymbol{x}',y')\sim\mathcal{B}_i^s}[\nabla_2 f_i]\}, \tag{9}$$

where only first-order derivatives of $f_i$ *w.r.t.* $\boldsymbol{\theta}_i$ are required.

### 3.4 CONVERGENCE ANALYSIS

Given notations, assumptions, and proofs in Appendix C, we have

**Theorem 1** (One-iteration deviation). *Let Assumption 1 to Assumption 3 hold. For an arbitrary client, after every communication iteration, we have*

$$\mathbb{E}[\mathcal{L}^{(t+1)E+1/2}] \leq \mathcal{L}^{tE+1/2} + (\frac{L_1\eta_c^2}{2} - \eta_c)\sum_{e=1/2}^{E-1} ||\nabla\mathcal{L}^{tE+e}||_2^2 + \frac{L_1 E \eta_c^2 \sigma^2}{2} + 2\eta_c^2\eta_s L_2 R' ER.$$

**Theorem 2** (Non-convex convergence rate of `FedL2G`). *Let Assumption 1 to Assumption 3 hold and $\Delta = \mathcal{L}^0 - \mathcal{L}^*$, where $\mathcal{L}^*$ refers to the local optimum. Given Theorem 1, for an arbitrary client and an arbitrary constant $\epsilon$, our `FedL2G` has a non-convex convergence rate $\mathcal{O}(1/T)$ with*

$$\frac{1}{T}\sum_{t=0}^{T-1}\sum_{e=1/2}^{E-1}\mathbb{E}[||\nabla\mathcal{L}^{tE+e}||_2^2] \leq \frac{\frac{2\Delta}{T} + L_1 E \eta_c^2 \sigma^2 + 4\eta_c^2\eta_s L_2 R' ER}{2\eta_c - L_1\eta_c^2} < \epsilon,$$

*where $0 < \eta_c < \frac{2\epsilon}{L_1(E\sigma^2+\epsilon)+4\eta_s L_2 R' ER}$ and $\eta_s > 0$.*

According to Theorem 2, our `FedL2G` can converge at a rate of $\mathcal{O}(1/T)$, and the server learning rate $\eta_s$ can be set to any positive value.

## 4 EXPERIMENTS

To evaluate the performance of our `FedL2G-l` and `FedL2G-f` alongside six popular *server-lightweight* HtFL methods: LG-FedAvg (Liang et al., 2020), FedGH (Yi et al., 2023), FML (Shen et al., 2020), FedKD (Wu et al., 2022), FedDistill (Jeong et al., 2018), and FedProto (Tan et al., 2022b), we conduct comprehensive experiments on four public datasets under two widely used

data heterogeneity settings, involving up to 14 heterogeneous model architectures. Specifically, we demonstrate the encouraging performance of FedL2G in accuracy, communication cost, and computation cost. Subsequently, we investigate the characteristics behind our FedL2G from an experimental perspective.

**Data heterogeneity settings.** Following existing work (Zhang et al., 2023d; Lin et al., 2020; Zhang et al., 2023b; 2024a), we adopt two popular settings across four enduring datasets Cifar10 (Krizhevsky & Geoffrey, 2009), Cifar100 (Krizhevsky & Geoffrey, 2009), Flowers102 (Nilsback & Zisserman, 2008), and Tiny-ImageNet (Chrabaszcz et al., 2017). Concretely, we simulate pathological data heterogeneity settings by allocating sub-datasets with 2/10/10/20 data classes from Cifar10/Cifar100/Flowers102/Tiny-ImageNet to each client. In Dirichlet data heterogeneity settings, we allocate the data of class $y$ to each client using a client-specific ratio $q^y$ from a given dataset. $q^y$ is sampled from a Dirichlet distribution with a control parameter $\beta$ as described in (Lin et al., 2020). By default, we set $\beta = 0.1$ for Cifar10 and Cifar100, and $\beta = 0.01$ for Flowers102 and Tiny-ImageNet to enhance setting diversity. In both the pathological and Dirichlet settings, the data quantity among clients varies to account for unbalanced scenarios.

**Model heterogeneity settings.** To neatly denote model heterogeneity settings, we utilize the notation HtFE$_X$ following the convention in (Zhang et al., 2024b) to represent a group of heterogeneous feature extractors, where $X$ denotes the degree of model heterogeneity (positive correlation), while the remaining classifier heads remain homogeneous. For example, HtFE$_8$ denotes a group of eight heterogeneous feature extractors from eight model architectures (4-layer CNN (McMahan et al., 2017), GoogleNet (Szegedy et al., 2015), MobileNet_v2 (Sandler et al., 2018), ResNet18, ResNet34, ResNet50, ResNet101, and ResNet152 (He et al., 2016)), respectively. In addition, we use the notation HtM$_X$ to denote a group of fully heterogeneous models. Within a specific group, for instance, HtFE$_X$, we allocate the $(i \mod X)$th model in this group to client $i$ with reinitialized parameters. Given the popularity of all models within HtFE$_8$ in the FL field, our primary focus is on utilizing HtFE$_8$. Additionally, some baseline methods, such as LG-FedAvg and FedGH, assume the classifier heads to be homogeneous, making HtM$_X$ inapplicable for them. Moreover, to meet the prerequisite of identical feature dimensions ($K$) for FedGH, FedKD, and FedProto, we incorporate an average pooling layer (Szegedy et al., 2015) before the classifier heads and set $K = 512$ for all models.

**Other necessary settings.** Following common practice (McMahan et al., 2017), we execute a complete local training epoch with a batch size of 10, i.e., $\lfloor \frac{n_i}{10} \rfloor$ update steps, during each communication iteration. We conduct each experiment for up to 1000 iterations across three trials, employing a client learning rate ($\eta_c$) of 0.01, and present the best results with error bars. Moreover, we examine full participation ($\rho = 1$), for 20 clients, while setting partial participation ($\rho = 0.5$) for scenarios involving 50 and 100 clients. Please refer to the Appendix A for more details and results.

## 4.1 ACCURACY IN TWO DATA HETEROGENEITY SETTINGS

Table 1: The test accuracy (%) on four datasets in two data heterogeneity settings using HtFE$_8$.

| Settings | Pathological Setting | | | | Dirichlet Setting | | | |
|---|---|---|---|---|---|---|---|---|
| Datasets | C10 | C100 | F102 | TINY | C10 | C100 | F102 | TINY |
| LG-FedAvg | 86.8±.3 | 57.0±.7 | 58.9±.3 | 32.0±.2 | 84.6±.5 | 40.7±.1 | 70.0±.9 | 48.2±.1 |
| FedGH | 86.6±.2 | 57.2±.2 | 59.3±.3 | 32.6±.4 | 84.4±.3 | 41.0±.5 | 69.7±.2 | 46.7±.1 |
| FML | 87.1±.2 | 55.2±.1 | 57.8±.3 | 31.4±.2 | 85.9±.1 | 39.9±.3 | 68.4±1.2 | 47.1±.1 |
| FedKD | 87.3±.3 | 56.6±.3 | 54.8±.4 | 32.6±.4 | 86.5±.2 | 40.6±.3 | 69.6±1.6 | 48.2±.5 |
| FedDistill | 87.2±.1 | 57.0±.3 | 58.5±.3 | 31.5±.4 | 86.0±.3 | 41.5±.1 | 71.2±.7 | 48.8±.1 |
| FedProto | 83.4±.2 | 53.6±.3 | 55.1±.2 | 29.3±.4 | 82.1±1.7 | 36.3±.3 | 62.3±.6 | 40.0±.1 |
| FedL2G-l | 87.7±.1 | 59.2±.4 | 60.3±.9 | 32.8±.7 | 86.5±.1 | 42.3±.1 | 71.5±.5 | 49.5±.3 |
| FedL2G-f | **89.3±.2** | **64.2±.3** | **64.2±.2** | **34.7±.3** | **87.6±.2** | **43.8±.4** | **73.6±.3** | **50.3±.4** |

To save space, we utilize brief abbreviations to represent the dataset names, specifically: "C10" for Cifar10, "C100" for Cifar100, "F102" for Flowers102, and "TINY" for Tiny-ImageNet. Based on Tab. 1, both FedL2G-l and FedL2G-f show superior performance compared to baseline methods.

Notably, `FedL2G-f` demonstrates better performance across all datasets and scenarios. This can be attributed to the fact that `FedL2G-l` learns to guide the original local task in the logit space, while `FedL2G-f` focuses on the intermediate feature space, and the latter contains richer information due to its higher dimension. Regarding accuracy, `FedL2G-f` surpasses the best baseline FedGH on Cifar100 by **7.0%** (a percentage improvement of 12.2%) in the pathological setting. Methods based on mutual distillation, such as FML and FedKD, transfer more information (with more bits) than other methods in each iteration. Yet, they do not consistently achieve optimal performance due to the absence of a teacher model with prior knowledge. FedProto suffers in the model heterogeneity setting and performs the worst, as client models exhibit varying feature extraction abilities (Zhang et al., 2024a). Conversely, our `FedL2G-f` excels with learning-to-guide in the intermediate feature space. While FedDistill mitigates this issue by sharing prototypical logits, there is still room for improvement through learning-to-guide in the logit space, a capability offered by `FedL2G-l`.

## 4.2 Accuracy in Additional Five Model Heterogeneity Settings

Table 2: The test accuracy (%) on Cifar100 in the default Dirichlet setting with incremental degrees of model heterogeneity or more clients.

| Settings | Incremental Degrees of Model Heterogeneity | | | | | More Clients ($\rho = 0.5$) | |
|---|---|---|---|---|---|---|---|
| | $HtFE_2$ | $HtFE_3$ | $HtFE_4$ | $HtFE_9$ | $HtM_{10}$ | $N = 50$ | $N = 100$ |
| LG-FedAvg | 46.6±.2 | 45.6±.4 | 43.9±.2 | 42.0±.3 | — | 37.8±.1 | 35.1±.5 |
| FedGH | 46.7±.4 | 45.2±.2 | 43.3±.2 | 43.0±.9 | — | 37.3±.4 | 34.3±.2 |
| FML | 45.9±.2 | 43.1±.1 | 43.0±.1 | 42.4±.3 | 39.9±.1 | 38.8±.1 | 36.1±.3 |
| FedKD | 46.3±.2 | 43.2±.5 | 43.2±.4 | 42.3±.4 | 40.4±.1 | 38.3±.4 | 35.6±.6 |
| FedDistill | 46.9±.1 | 43.5±.2 | 43.6±.1 | 42.1±.2 | 41.0±.1 | 38.5±.4 | 36.1±.2 |
| FedProto | 44.0±.2 | 38.1±.6 | 34.7±.6 | 32.7±.8 | 36.1±.1 | 33.0±.4 | 29.0±.5 |
| `FedL2G-l` | 47.3±.1 | 44.5±.1 | **44.8±.1** | 44.1±.1 | 41.8±.2 | 38.9±.2 | 36.7±.1 |
| `FedL2G-f` | **47.8±.3** | **45.8±.1** | 44.7±.1 | **45.7±.2** | **43.5±.1** | **40.5±.0** | **37.9±.3** |

Besides the $HtFE_8$ group, we also explore five other model heterogeneity settings, while maintaining consistent data heterogeneity in the Dirichlet setting to control variables. The degree of model heterogeneity escalates from $HtFE_2$ to $HtM_{10}$ as follows: $HtFE_2$ comprises 4-layer CNN and ResNet18; $HtFE_3$ includes ResNet10 (Zhong et al., 2017), ResNet18, and ResNet34; $HtFE_4$ comprises 4-layer CNN, GoogleNet, MobileNet_v2, and ResNet18; $HtFE_9$ includes ResNet4, ResNet6, and ResNet8 (Zhong et al., 2017), ResNet10, ResNet18, ResNet34, ResNet50, ResNet101, and ResNet152; $HtM_{10}$ contains all the model architectures in $HtFE_8$ plus two additional architectures ViT-B/16 (Dosovitskiy et al., 2020) and ViT-B/32 (Dosovitskiy et al., 2020). ViT models have a complex classifier head, whereas other CNN-based models only consider the last fully connected layer as the classifier head. Consequently, methods assuming a homogeneous classifier head, such as LG-FedAvg and FedGH, do not apply to $HtM_{10}$. Referring to Tab. 2, our `FedL2G-l` and `FedL2G-f` still perform well in these scenarios, particularly in more model-heterogeneous settings. As the setting becomes more heterogeneous, finding consistent knowledge to share becomes increasingly challenging, and negative transfer (Cui et al., 2022) may also arise. However, learning-to-guide knowledge is generic, making it easy for `FedL2G` to aggregate and distribute this knowledge in diverse scenarios, benefiting all clients.

## 4.3 Accuracy With More Clients or More Local Training Epochs

**More Clients.** In addition to experimenting with a total of 20 clients, we extend our evaluation by incorporating more clients created using the given Cifar100 dataset. With an increase in the number of clients, maintaining a consistent total data amount across all clients results in less local data on each client. In these scenarios, with a partial client participation ratio of $\rho = 0.5$, our `FedL2G-l` and `FedL2G-f` can still maintain their superiority, as shown in Tab. 2.

**More Local Training Epochs.** Increasing the number of local epochs, denoted by $E$, in each communication iteration can reduce the total number of iterations required for convergence, consequently lowering total communication overhead (McMahan et al., 2017; Zhang et al., 2024b). In

Tab. 3, FedGH experiences approximately a 1% decrease in accuracy when $E \geq 10$. Since the globally shared model struggles with data heterogeneity, FML and FedKD also exhibit performance degradation with a larger $E$, albeit more severe. Specifically, FML and FedKD continue to decrease from $E = 5$ to $E = 20$, with FML dropping by 3.1% and FedKD dropping by 2.0%. In contrast, our `FedL2G-l` and `FedL2G-f` consistently uphold superior performance even with a larger $E$. Remarkably, `FedL2G-f` shows an increase of 0.6% in accuracy from $E = 5$ to $E = 20$, showcasing its exceptional adaptability in scenarios with low communication quality.

Table 3: The test accuracy (%) on Cifar100 in the default Dirichlet setting using HtFE$_8$ for three experiments. "MB" and "s" are short for megabyte and second, respectively. The time in the brackets represents the cost of the warm-up period, several times less than local training.

| Experiments | Local Training Epochs | | | Comm. (MB) | | Computation (s) | |
|---|---|---|---|---|---|---|---|
| | $E = 5$ | $E = 10$ | $E = 20$ | Up. | Down. | Client | Server |
| LG-FedAvg | 40.3±.2 | 40.5±.1 | 40.9±.2 | 3.93 | 3.93 | 6.18 | 0.04 |
| FedGH | 41.1±.3 | 39.9±.3 | 40.2±.4 | 1.75 | 3.93 | 9.53 | 0.37 |
| FML | 39.1±.3 | 38.0±.2 | 36.0±.2 | 70.57 | 70.57 | 8.63 | 0.07 |
| FedKD | 41.1±.1 | 40.4±.2 | 39.1±.3 | 63.02 | 63.02 | 9.04 | 0.07 |
| FedDistill | 41.0±.3 | 41.3±.2 | 41.1±.4 | 0.34 | 0.76 | 6.52 | 0.03 |
| FedProto | 38.0±.5 | 38.1±.4 | 38.7±.5 | 1.75 | 3.89 | 6.65 | 0.04 |
| `FedL2G-l` | 42.2±.2 | 42.0±.2 | 42.1±.1 | 0.34 | 0.76 | 7.49 (2.23) | 0.03 |
| `FedL2G-f` | **43.7±.1** | **43.8±.2** | **44.3±.3** | 1.75 | 3.89 | 8.84 (2.24) | 0.04 |

## 4.4 COMMUNICATION AND COMPUTATION OVERHEAD

**Communication cost.** We consider both the upload and download bytes (across all participating clients) as part of the communication overhead in each iteration, using a float32 (= 4 bytes) data type in PyTorch (Paszke et al., 2019) to store each floating number. In Tab. 3, despite FML and FedKD transmitting a relatively small global model, their communication costs remain significantly high compared to other methods that share lightweight components. The SVD technique in FedKD (Wu et al., 2022), does not significantly reduce the communication overhead. Given that we only upload the gradients of guiding vectors on the client, the communication cost of `FedL2G-l` and `FedL2G-f` is equivalent to that of FedDistill and FedProto, respectively. This cost falls within the lowest group among these methods.

**Computation cost.** To capture essential operations, we measure the averaged GPU execution time of each client and the server on an idle GPU card in each iteration and show the time cost in Tab. 3. As FedGH gathers prototypes after local training, it costs extra time for inferencing across the entire training set using the trained client model. In contrast, FedDistill and FedProto collect prototypical logits and features, respectively, concurrently with model training in each batch, thereby eliminating this additional cost. Besides, FedGH trains the global head on the server consuming relatively more power, even with one server epoch per iteration. Since we only average gradients on the server and update $\mathcal{G}$ once without backpropagation, our `FedL2G-l` and `FedL2G-f` demonstrate similar time-efficiency to FedDistill and FedProto, respectively. Due to the extra learning-to-guide process, `FedL2G` costs more client time than FedDistill and FedProto. However, `FedL2G-l` still requires less time than FML, FedKD, and FedGH, and the improved test accuracy justifies this cost.

## 4.5 FEDL2G PRIORITIZES THE ORIGINAL TASK

Beyond presenting the test accuracy, we examine the training losses by examining the intrinsic training process. For each method, we illustrate only the original local loss, i.e., $\ell_{ce}$, in Fig. 3. Specifically, we aggregate all the clients' original local losses through weighted averaging, following Eq. (1). These original local loss curves closely align with the accuracy trends in Tab. 2 (HtFE$_9$), indicating that lower original local loss corresponds to higher test accuracy in our scenarios. Since our `FedL2G` learns guiding vectors that help the client model focus more on its original task, `FedL2G-l` and `FedL2G-f` achieve the second-lowest and lowest losses, respectively.

Besides the magnitude of the original local losses, our `FedL2G` method also offers advantages in smoothness and convergence speed. From Fig. 3, we observe that the loss curves of FedDistill, FedGH, FedProto, and LG-FedAvg fluctuate significantly in the beginning. The growth of $\ell_{ce}$ can be attributed to the mismatch of the shared global information and clients' tasks. Given that `FedL2G-l` and `FedL2G-f` focus on clients' original tasks, we can introduce more client-required information for guiding vectors, leading to a stable reduction in the original local loss. Because of the same benefits, `FedL2G-f` can converge at a relatively early iteration and achieve the highest test accuracy simultaneously. Despite the lesser amount of guiding information in `FedL2G-l` compared to `FedL2G-f`, `FedL2G-l` also demonstrates superiority in terms of smoothness and convergence when compared to FedDistill.

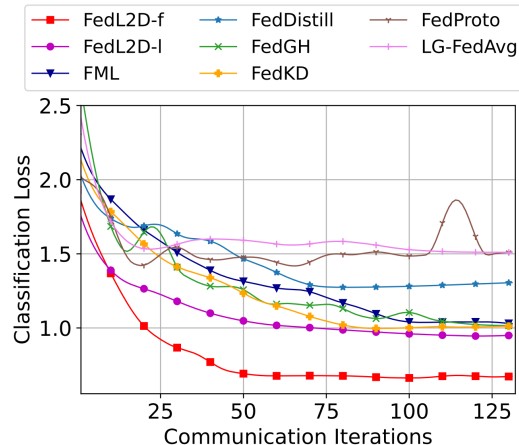

Figure 3: The averaged original local loss ($\ell_{ce}$) of all clients for nine HtFL methods on Cifar100 in the default Dirichlet setting using HtFE$_9$.

### 4.6 FEDL2G PROTECTS FEATURE INFORMATION

Differing from FedDistill and FedProto, which gather data-derived prototypical logits and features from the clients, we collect the gradients of randomly initialized guiding vectors. These gradients are calculated using a complex formula (refer to Eq. (7)) to reduce the original local losses for all clients. Therefore, our `FedL2G` does not directly upload client data-related information and safeguards the feature information for clients. In a sense, logit vectors are also feature vectors with lower dimensions. Here, we illustrate the t-SNE (Van der Maaten & Hinton, 2008) visualization of the global prototypes $\{g^y\}_{y=1}^{C}$ (obtained via Eq. (2)) and the guiding vectors $\{v^y\}_{y=1}^{C}$ from `FedL2G-l` and `FedL2G-f`. As per Fig. 4, guiding vectors differ from global prototypes because they do not

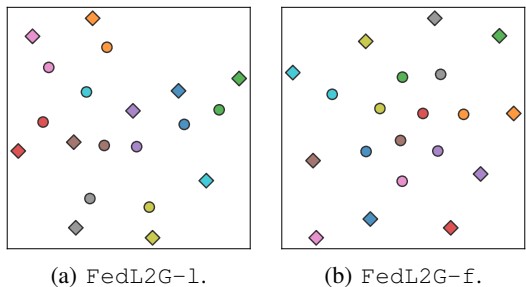

(a) `FedL2G-l`.   (b) `FedL2G-f`.

Figure 4: The t-SNE visualization of guiding vectors (diamonds) and feature vectors (circles) on Cifar10 in the default Dirichlet setting using HtFE$_8$. Different colors represent different classes. *Best viewed in color.*

overlap. Moreover, guiding vectors and global prototypes of the same class do not always cluster. Instead, guiding vectors and global prototypes from different classes can be closer, providing additional protection for the class information of local features. This phenomenon is more pronounced in `FedL2G-f`, where the distances between guiding vectors and global prototypes are larger than in `FedL2G-l`. This is because the guiding vectors in `FedL2G-f` have relatively more parameters and knowledge to learn. Given that a larger distance signifies improved discrimination and guidance for the class-wise vectors utilized in a guiding loss (Zhang et al., 2024a), our guiding vectors exhibit greater separability than the global prototypes, indicating enhanced guidance capability for the client models.

## 5 CONCLUSION

We observe the original local loss growth phenomenon on the client in prior prototype-based HtFL methods when guided by global prototypes. Then we attribute this problem to the mismatch between the guiding objective and the client's original local objective. To address this issue, we propose a `FedL2G` approach to reduce the client's original objective when using guiding vectors by prioritizing the local objective during the learning of guiding vectors. The superiority of `FedL2G` is evidenced through theoretical analysis and extensive experiments.

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

## A  ADDITIONAL EXPERIMENTAL DETAILS

**Datasets and environment.**   We use four datasets with their respective download links: Cifar10[2], Cifar100[3], Flowers102[4], and Tiny-ImageNet[5]. We split all client data into a training set and a test set for each client at a ratio of 75% and 25%, respectively, and we evaluate the averaged test accuracy on clients' test sets. All our experiments are conducted on a machine with 64 Intel(R) Xeon(R) Platinum 8362 CPUs, 256G memory, eight NVIDIA 3090 GPUs, and Ubuntu 20.04.4 LTS. Most of our experiments can be completed within 48 hours, while others, involving many clients and extensive local training epochs, may take up to a week to finish.

**Hyperparameter settings.**   For our baseline methods, we set their hyperparameters following existing work (Zhang et al., 2024a;b). As for our `FedL2G-l` and `FedL2G-f`, we tune the server learning rate $\eta_s$ (the *unique* hyperparameter) by grid search on the Cifar100 dataset in the default Dirichlet setting with HtFE$_8$ and use an identical setting on all experimental tasks without further tuning. Specifically, we search $\eta_s$ in the range: $\{0.01, 0.05, 0.1, 0.5, 1, 10, 50, 100, 500\}$. We set $\eta_s = 0.1$ for `FedL2G-l` and set $\eta_s = 100$ for `FedL2G-f`. The $\eta_s$ hyperparameters of `FedL2G-l` and `FedL2G-f` differ due to their discrepancy in the learnable knowledge capacity of the guiding vectors. The dimension of the guiding vectors in `FedL2G-f` is larger than in `FedL2G-l`, necessitating more server updates.

**The small auxiliary model for FML and FedKD.**   As FML and FedKD utilize a global auxiliary model for mutual distillation, this auxiliary model needs to be as compact as possible to minimize communication overhead during model parameter transmission (Wu et al., 2022). Therefore, we opt for the smallest model within each group of heterogeneous models to serve as the auxiliary model in all scenarios.

## B  SENSITIVITY STUDY

We conduct a sensitivity study here to further study the influence of the server learning rate $\eta_s$. From Tab. 4, we know that `FedL2G-l` and `FedL2G-f` benefit from distinct ranges of $\eta_s$, also attributed to their different trainable parameters and learning capacities. Moreover, `FedL2G-f` demonstrates higher optimal accuracy than `FedL2G-l`, while `FedL2G-l` yields a more stable outcome across different $\eta_s$.

|  | $\eta_s = 0.01$ | $\eta_s = 0.05$ | $\eta_s = 0.1$ | $\eta_s = 0.5$ | $\eta_s = 1$ |
|---|---|---|---|---|---|
| `FedL2G-l` | 41.7±.3 | 41.6±.1 | **42.3±.1** | 41.6±.5 | 41.8±.3 |
|  | $\eta_s = 1$ | $\eta_s = 10$ | $\eta_s = 50$ | $\eta_s = 100$ | $\eta_s = 500$ |
| `FedL2G-f` | 41.1±.5 | 42.0±.1 | 43.5±.1 | **43.8±.4** | 41.4±.5 |

Table 4: The test accuracy (%) of `FedL2G-l` and `FedL2G-f` on Cifar100 in the default Dirichlet setting using HtFE$_8$ with different $\eta_s$.

## C  THEORETICAL ANALYSIS

Here we bring some existing equations for convenience. Recall that we have $N$ clients training their heterogeneous local models (with parameters $\boldsymbol{\theta}_1, \ldots, \boldsymbol{\theta}_N$) using their private and heterogeneous training data $\mathcal{D}_1, \ldots, \mathcal{D}_N$. Besides, they share global guiding vectors $\mathcal{G} = \{\boldsymbol{v}^y\}_{y=1}^C$, with the

---

[2]`https://pytorch.org/vision/main/generated/torchvision.datasets.CIFAR10.html`

[3]`https://pytorch.org/vision/stable/generated/torchvision.datasets.CIFAR100.html`

[4]`https://pytorch.org/vision/stable/generated/torchvision.datasets.Flowers102.html`

[5]`http://cs231n.stanford.edu/tiny-imagenet-200.zip`

assistance of a server to facilitate collaborative learning. Formally, the objective of `FedL2G` is

$$\min_{\boldsymbol{\theta}_1,\dots,\boldsymbol{\theta}_N} \sum_{i=1}^{N} \frac{|\mathcal{D}_i|}{D} \mathcal{L}_{\mathcal{D}_i}(\boldsymbol{\theta}_i, \mathcal{G}), \tag{C.1}$$

where the total client loss $\mathcal{L}_{\mathcal{D}_i}$ is defined by

$$\mathcal{L}_{\mathcal{D}_i}(\boldsymbol{\theta}_i, \mathcal{G}) := \mathbb{E}_{(\boldsymbol{x},y)\sim\mathcal{D}_i}[\ell_{ce}(f_i(\boldsymbol{x}, \boldsymbol{\theta}_i), y) + \ell_g(f_i(\boldsymbol{x}, \boldsymbol{\theta}_i), \boldsymbol{v}^y)], \tag{C.2}$$

and the original local loss $\mathcal{L}'_{\mathcal{D}_i}$ is defined by

$$\mathcal{L}'_{\mathcal{D}_i}(\boldsymbol{\theta}_i, \mathcal{G}) := \mathbb{E}_{(\boldsymbol{x},y)\sim\mathcal{D}_i}[\ell_{ce}(f_i(\boldsymbol{x}, \boldsymbol{\theta}_i), y)]. \tag{C.3}$$

Here we consider `FedL2G-l` for simplicity, and it is easy to extend theoretical analysis to `FedL2G-f` by substituting $\ell_g(f_i(\boldsymbol{x}, \boldsymbol{\theta}_i), \boldsymbol{v}^y)$ with $\ell_g(h_i(\boldsymbol{x}, \boldsymbol{\theta}_i^h), \boldsymbol{v}^y)$. We optimize global $\mathcal{G}$ by

$$\mathcal{G}^t = \mathcal{G}^{t-1} - \eta_s \frac{1}{N} \sum_{i\in[N]} \nabla_{\mathcal{G}^{t-1}} \mathbb{E}_{(\boldsymbol{x},y)\sim\mathcal{D}_i^q}[\ell_{ce}(f_i(\boldsymbol{x}, \boldsymbol{\theta}_i - \eta_c \nabla_{\boldsymbol{\theta}_i}\mathcal{L}_{\mathcal{B}_i^s}(\boldsymbol{\theta}_i, \mathcal{G}^{t-1})), y)], \tag{C.4}$$

where we consider full participation for simplicity. The convergence of Eq. (C.2) for any client is equivalent to the convergence of `FedL2G`'s objective in Eq. (C.1). Thus, in the following, we omit the client notation $i$ and some corresponding notations, such as $\mathcal{D}_i$.

To further examine the local training process, in addition to the communication iteration notation $t$, we introduce $e \in \{1/2, 1, 2, \dots, E\}$ to represent the local step. We denote the $e$th local training step in iteration $t$ as $tE + e$. Specifically, $tE + 1/2$ refers to the moment when clients receive $\mathcal{G}$ before local training. We adopt four assumptions, partially based on FedProto (Tan et al., 2022b).

**Assumption 1** (Unbiased Gradient and Bounded Variance). *The stochastic gradient $\omega^t = \nabla\mathcal{L}_\xi(\boldsymbol{\theta}^t, \mathcal{G}^t)$ is an unbiased estimation of each client's gradient w.r.t. its loss:*

$$\mathbb{E}_{\xi\sim\mathcal{D}}[\omega^t] = \nabla\mathcal{L}(\boldsymbol{\theta}^t, \mathcal{G}) = \nabla\mathcal{L}^t.$$

*and its variance is bounded by $\sigma^2$:*

$$\mathbb{E}[||\omega^t - \nabla\mathcal{L}^t||_2^2] \leq \sigma^2.$$

**Assumption 2** (Bounded Gradient). *The expectation of the stochastic gradient $\omega^t$ and $\omega'^t = \nabla\mathcal{L}'_\xi(\boldsymbol{\theta}^t, \mathcal{G}^t)$ are bounded by $R$ and $R'$, respectively:*

$$\mathbb{E}[||\omega^t||_2] \leq R, \quad \mathbb{E}[||\omega'^t||_2] \leq R'.$$

**Assumption 3** (Lipschitz Smoothness). *Each total local objective $\mathcal{L}$ is $L_1$-Lipschitz smooth, which also means the gradient of $\mathcal{L}$ is $L_1$-Lipschitz continuous, i.e.,*

$$||\nabla\mathcal{L}^{t_1} - \nabla\mathcal{L}^{t_2}||_2 \leq L_1||\boldsymbol{\theta}^{t_1} - \boldsymbol{\theta}^{t_2}||_2, \quad \forall t_1, t_2 > 0,$$

*which implies the following quadratic bound,*

$$\mathcal{L}^{t_1} - \mathcal{L}^{t_2} \leq \langle\nabla\mathcal{L}^{t_2}, (\boldsymbol{\theta}^{t_1} - \boldsymbol{\theta}^{t_2})\rangle + \frac{1}{2}L_1||\boldsymbol{\theta}^{t_1} - \boldsymbol{\theta}^{t_2}||_2^2, \quad \forall t_1, t_2 > 0.$$

*Besides, each client model function $f$ is $L_2$-Lipschitz smooth, i.e.,*

$$||\nabla f^{t_1} - \nabla f^{t_2}||_2 \leq L_2||\boldsymbol{\theta}^{t_1} - \boldsymbol{\theta}^{t_2}||_2, \quad \forall t_1, t_2 > 0.$$

Given Assumption 1 and Assumption 2, any client's gradient w.r.t. $\mathcal{G}$ is

$$\pi^{t-1} = \nabla_{\mathcal{G}^{t-1}} \mathbb{E}_{(\boldsymbol{x},y)\sim\mathcal{D}^q}[\ell_{ce}(f(\boldsymbol{x}, \boldsymbol{\theta} - \eta_c\nabla_{\boldsymbol{\theta}}\mathcal{L}_{\mathcal{B}^s}(\boldsymbol{\theta}, \mathcal{G}^{t-1})), y)] \tag{C.5}$$

$$= \mathbb{E}_{(\boldsymbol{x},y)\sim\mathcal{D}^q}[\nabla_{\mathcal{G}^{t-1}}\ell_{ce}(f(\boldsymbol{x}, \boldsymbol{\theta} - \eta_c\nabla_{\boldsymbol{\theta}}\mathcal{L}_{\mathcal{B}^s}(\boldsymbol{\theta}, \mathcal{G}^{t-1})), y)] \tag{C.6}$$

$$= \mathbb{E}_{(\boldsymbol{x},y)\sim\mathcal{D}^q}[\nabla_1\ell_{ce} \cdot \nabla_2 f \cdot \nabla_{\mathcal{G}^{t-1}}(\boldsymbol{\theta} - \eta_c\nabla_{\boldsymbol{\theta}}\mathcal{L}_{\mathcal{B}^s}(\boldsymbol{\theta}, \mathcal{G}^{t-1}))] \tag{C.7}$$

$$= -\eta_c\mathbb{E}_{(\boldsymbol{x},y)\sim\mathcal{D}^q}[\nabla_1\ell_{ce} \cdot \nabla_2 f \cdot \nabla_{\mathcal{G}^{t-1}}\nabla_{\boldsymbol{\theta}}\mathcal{L}_{\mathcal{B}^s}(\boldsymbol{\theta}, \mathcal{G}^{t-1})] \tag{C.8}$$

$$= -\eta_c\mathbb{E}_{(\boldsymbol{x},y)\sim\mathcal{D}^q}\{\nabla_1\ell_{ce} \cdot \nabla_2 f \cdot \mathbb{E}_{(\boldsymbol{x}',y')\sim\mathcal{B}^s}[\nabla_{\mathcal{G}^{t-1}}\nabla_{\boldsymbol{\theta}}\ell_g(f(\boldsymbol{x}', \boldsymbol{\theta}), \boldsymbol{v}^{y'})]\} \tag{C.9}$$

$$= -\eta_c\mathbb{E}_{(\boldsymbol{x},y)\sim\mathcal{D}^q}\{\nabla_1\ell_{ce} \cdot \nabla_2 f \cdot \mathbb{E}_{(\boldsymbol{x}',y')\sim\mathcal{B}^s}[\nabla_2 f \cdot \nabla_{\mathcal{G}^{t-1}}\nabla_1\ell_g]\} \tag{C.10}$$

$$= 2\eta_c\mathbb{E}_{(\boldsymbol{x},y)\sim\mathcal{D}^q}\{\nabla_1\ell_{ce} \cdot \nabla_2 f \cdot \mathbb{E}_{(\boldsymbol{x}',y')\sim\mathcal{B}^s}[\nabla_2 f]\}, \tag{C.11}$$

where $\nabla_1 \ell_{ce} := \nabla_{a_1} \ell_{ce}(a_1, a_2)$, indicating the derivative of $\ell_{ce}$ *w.r.t.* the first variable, and so for $\nabla_2 f$ and $\nabla_1 \ell_g$. Under Assumption 1, we can mimic regular training through the pseudo-train step ③, as $\mathcal{B}^s$ is randomly re-sampled in each iteration. All the derivatives in Eq. (C.11) are bounded under Assumption 2.

Then, we have two key lemmas:

**Lemma 1.** *Let Assumption 1 and Assumption 3 hold. The total client loss of an arbitrary client can be bounded:*

$$\mathbb{E}[\mathcal{L}^{(t+1)E}] \leq \mathcal{L}^{tE+1/2} + (\frac{L_1 \eta_c^2}{2} - \eta_c) \sum_{e=1/2}^{E-1} ||\nabla \mathcal{L}^{tE+e}||_2^2 + \frac{L_1 E \eta_c^2 \sigma^2}{2}.$$

*Proof.* This lemma focuses solely on local training at the client level, incorporating both the original local objective and the guiding objective. It can be easily derived by substituting the relevant notations from Lemma 1 of the prototype-based HtFL method, FedProto. □

**Lemma 2.** *Let Assumption 2 and Assumption 3 hold. After the guiding vectors are updated on the server and downloaded to clients, the total client loss of an arbitrary client can be bounded:*

$$\mathbb{E}[\mathcal{L}^{(t+1)E+1/2}] \leq \mathcal{L}^{(t+1)E} + 2\eta_c^2 \eta_s L_2 R' E R.$$

*Proof.*

$$\mathcal{L}^{(t+1)E+1/2} = \mathcal{L}^{(t+1)E} + \mathcal{L}^{(t+1)E+1/2} - \mathcal{L}^{(t+1)E} \tag{C.12}$$

$$= \mathcal{L}^{(t+1)E} + ||f(\boldsymbol{\theta}^{(t+1)E}) - \mathcal{G}^{(t+2)E}||_2 - ||f(\boldsymbol{\theta}^{(t+1)E}) - \mathcal{G}^{(t+1)E}||_2 \tag{C.13}$$

$$\overset{(a)}{\leq} \mathcal{L}^{(t+1)E} + ||\mathcal{G}^{(t+2)E} - \mathcal{G}^{(t+1)E}||_2 \tag{C.14}$$

$$= \mathcal{L}^{(t+1)E} + \eta_s ||\mathbb{E}_{[N]}(\pi^{(t+1)E} - \pi^{(t+2)E})||_2 \tag{C.15}$$

$$\overset{(b)}{\leq} \mathcal{L}^{(t+1)E} + \eta_s \mathbb{E}_{[N]} ||\pi^{(t+1)E} - \pi^{(t+2)E}||_2 \tag{C.16}$$

$$\overset{(c)}{\leq} \mathcal{L}^{(t+1)E} + 2\eta_c \eta_s \mathbb{E}_{[N]} \mathbb{E}_{\mathcal{D}} ||\nabla_1 \ell_{ce}^{(t+1)E} \cdot \nabla_2 f^{(t+1)E} \cdot \mathbb{E}_{\xi}[\nabla_2 f^{(t+1)E}] - \nabla_1 \ell_{ce}^{tE} \cdot \nabla_2 f^{tE} \cdot \mathbb{E}_{\xi}[\nabla_2 f^{tE}]||_2 \tag{C.17}$$

$$\overset{(d)}{\leq} \mathcal{L}^{(t+1)E} + 2\eta_c \eta_s R' \mathbb{E}_{[N]} \mathbb{E}_{\xi} ||\nabla_2 f^{(t+1)E} - \nabla_2 f^{tE}||_2 \tag{C.18}$$

$$\overset{(e)}{\leq} \mathcal{L}^{(t+1)E} + 2\eta_c \eta_s L_2 R' \mathbb{E}_{[N]} \mathbb{E}_{\xi} ||\boldsymbol{\theta}^{(t+1)E} - \boldsymbol{\theta}^{tE}||_2 \tag{C.19}$$

$$\overset{(f)}{\leq} \mathcal{L}^{(t+1)E} + 2\eta_c^2 \eta_s L_2 R' \mathbb{E}_{[N]} \mathbb{E}_{\xi} \sum_{e=1/2}^{E-1} ||\omega^{tE+e}||_2 \tag{C.20}$$

Take expectations of random variable $\xi$, we have

$$\mathbb{E}[\mathcal{L}^{(t+1)E+1/2}] \leq \mathcal{L}^{(t+1)E} + 2\eta_c^2 \eta_s L_2 R' \mathbb{E}_{[N]} \mathbb{E}_{\xi} \sum_{e=1/2}^{E-1} ||\omega^{tE+e}||_2 \tag{C.21}$$

$$\overset{(g)}{\leq} \mathcal{L}^{(t+1)E} + 2\eta_c^2 \eta_s L_2 R' E R. \tag{C.22}$$

In the above inequations, (a) follows from $||a - b||_2 - ||a - c||_2 \leq ||b - c||_2$; (b), (c), and (f) follow from $||\sum a_j||_2 \leq \sum ||a_j||_2$, where $\mathbb{E}_{\mathcal{D}} a$ denotes taking expectations of $a$ over set $\mathcal{D}$, *e.g.*, $\mathbb{E}_{[N]} a$ means $\mathbb{E}_{i \sim \{1,...,N\}} a_j$; (d) follows from Assumption 1 and Assumption 2, where $\mathcal{L}'(\boldsymbol{\theta}, \mathcal{G}) = \nabla_1 \ell_{ce} \cdot \nabla_2 f$; (e) follows from Assumption 3; (g) follows from Assumption 2. □

Then, we have

**Theorem 1** (One-iteration deviation). *Let Assumption 1 to Assumption 3 hold. For an arbitrary client, after every communication iteration, we have*

$$\mathbb{E}[\mathcal{L}^{(t+1)E+1/2}] \leq \mathcal{L}^{tE+1/2} + (\frac{L_1\eta_c^2}{2} - \eta_c)\sum_{e=1/2}^{E-1}||\nabla\mathcal{L}^{tE+e}||_2^2 + \frac{L_1 E\eta_c^2\sigma^2}{2} + 2\eta_c^2\eta_s L_2 R' ER.$$

*Proof.* Taking expectation of $\boldsymbol{\theta}$ on both sides in Lemma 2, we have

$$\mathbb{E}[\mathcal{L}^{(t+1)E+1/2}] \leq \mathbb{E}[\mathcal{L}^{(t+1)E}] + 2\eta_c^2\eta_s L_2 R' ER. \tag{C.23}$$

Then summing Eq. (C.23) and Lemma 1 up, we have

$$\mathbb{E}[\mathcal{L}^{(t+1)E+1/2}] \leq \mathcal{L}^{tE+1/2} + (\frac{L_1\eta_c^2}{2} - \eta_c)\sum_{e=1/2}^{E-1}||\nabla\mathcal{L}^{tE+e}||_2^2 + \frac{L_1 E\eta_c^2\sigma^2}{2} + 2\eta_c^2\eta_s L_2 R' ER.$$

$$\tag{C.24}$$

$\square$

**Theorem 2** (Non-convex convergence rate of `FedL2G`). *Let Assumption 1 to Assumption 3 hold and $\Delta = \mathcal{L}^0 - \mathcal{L}^*$, where $\mathcal{L}^*$ refers to the local optimum. Given Theorem 1, for an arbitrary client and an arbitrary constant $\epsilon$, our `FedL2G` has a non-convex convergence rate $\mathcal{O}(1/T)$ with*

$$\frac{1}{T}\sum_{t=0}^{T-1}\sum_{e=1/2}^{E-1}\mathbb{E}[||\nabla\mathcal{L}^{tE+e}||_2^2] \leq \frac{\frac{2\Delta}{T} + L_1 E\eta_c^2\sigma^2 + 4\eta_c^2\eta_s L_2 R' ER}{2\eta_c - L_1\eta_c^2} < \epsilon,$$

*where $0 < \eta_c < \frac{2\epsilon}{L_1(E\sigma^2+\epsilon)+4\eta_s L_2 R' ER}$ and $\eta_s > 0$.*

*Proof.* By interchanging the left and right sides of Eq. (C.24), we can get

$$\sum_{e=1/2}^{E-1}||\nabla\mathcal{L}^{tE+e}||_2^2 \leq \frac{\mathcal{L}^{tE+1/2} - \mathbb{E}[\mathcal{L}^{(t+1)E+1/2}] + \frac{L_1 E\eta_c^2\sigma^2}{2} + 2\eta_c^2\eta_s L_2 R' ER}{\eta_c - \frac{L_1\eta_c^2}{2}}, \tag{C.25}$$

when $\eta_c - \frac{L_1\eta_c^2}{2} > 0$, *i.e.*, $0 < \eta_c < \frac{2}{L_1}$. Taking the expectation of $\boldsymbol{\theta}$ on both sides and summing all inequalities overall communication iterations, we obtain

$$\frac{1}{T}\sum_{t=0}^{T-1}\sum_{e=1/2}^{E-1}\mathbb{E}[||\nabla\mathcal{L}^{tE+e}||_2^2] \leq \frac{\frac{1}{T}\sum_{t=0}^{T-1}(\mathcal{L}^{tE+1/2} - \mathbb{E}[\mathcal{L}^{(t+1)E+1/2}]) + \frac{L_1 E\eta_c^2\sigma^2}{2} + 2\eta_c^2\eta_s L_2 R' ER}{\eta_c - \frac{L_1\eta_c^2}{2}}.$$

$$\tag{C.26}$$

Let $\Delta = \mathcal{L}^0 - \mathcal{L}^* > 0$, we have $\frac{1}{T}\sum_{t=0}^{T-1}(\mathcal{L}^{tE+1/2} - \mathbb{E}[\mathcal{L}^{(t+1)E+1/2}]) \leq \Delta$ and

$$\frac{1}{T}\sum_{t=0}^{T-1}\sum_{e=1/2}^{E-1}\mathbb{E}[||\nabla\mathcal{L}^{tE+e}||_2^2] \leq \frac{\frac{2\Delta}{T} + L_1 E\eta_c^2\sigma^2 + 4\eta_c^2\eta_s L_2 R' ER}{2\eta_c - L_1\eta_c^2}. \tag{C.27}$$

Given any $\epsilon > 0$, let

$$\frac{\frac{2\Delta}{T} + L_1 E\eta_c^2\sigma^2 + 4\eta_c^2\eta_s L_2 R' ER}{2\eta_c - L_1\eta_c^2} < \epsilon, \tag{C.28}$$

we have

$$T > \frac{2\Delta}{\epsilon\eta_c(2 - L_1\eta_c) - \eta_c^2(L_1 E\sigma^2 + 4\eta_s L_2 R' ER)}. \tag{C.29}$$

In this context, we have

$$\frac{1}{T}\sum_{t=0}^{T-1}\sum_{e=1/2}^{E-1}\mathbb{E}[||\nabla\mathcal{L}^{tE+e}||_2^2] \leq \epsilon, \tag{C.30}$$

when

$$0 < \eta_c < \frac{2\epsilon}{L_1(E\sigma^2 + \epsilon) + 4\eta_s L_2 R' ER} < \frac{2}{L_1}, \tag{C.31}$$

and

$$\eta_s > 0 \tag{C.32}$$

Since all the notations of the right side in Eq. (C.27) are given constants except for $T$, our `FedL2G`'s non-convex convergence rate is $\epsilon \sim \mathcal{O}(1/T)$. $\square$

# D  VISUALIZATIONS OF DATA DISTRIBUTIONS

We illustrate the data distributions on all clients in the above experiments in the following.

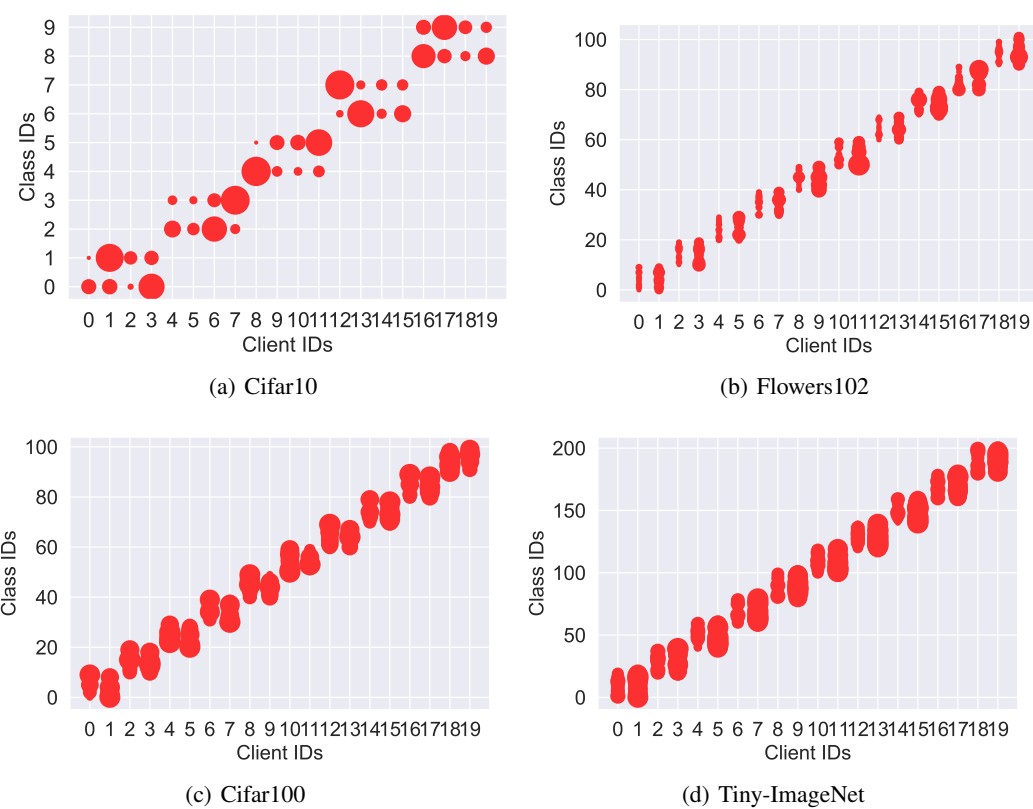

| (a) Cifar10 | (b) Flowers102 |
|---|---|
| (c) Cifar100 | (d) Tiny-ImageNet |

Figure 5: The data distribution of each client on Cifar10, Flowers102, Cifar100, and Tiny-ImageNet, respectively, in the pathological settings. The size of a circle represents the number of samples.

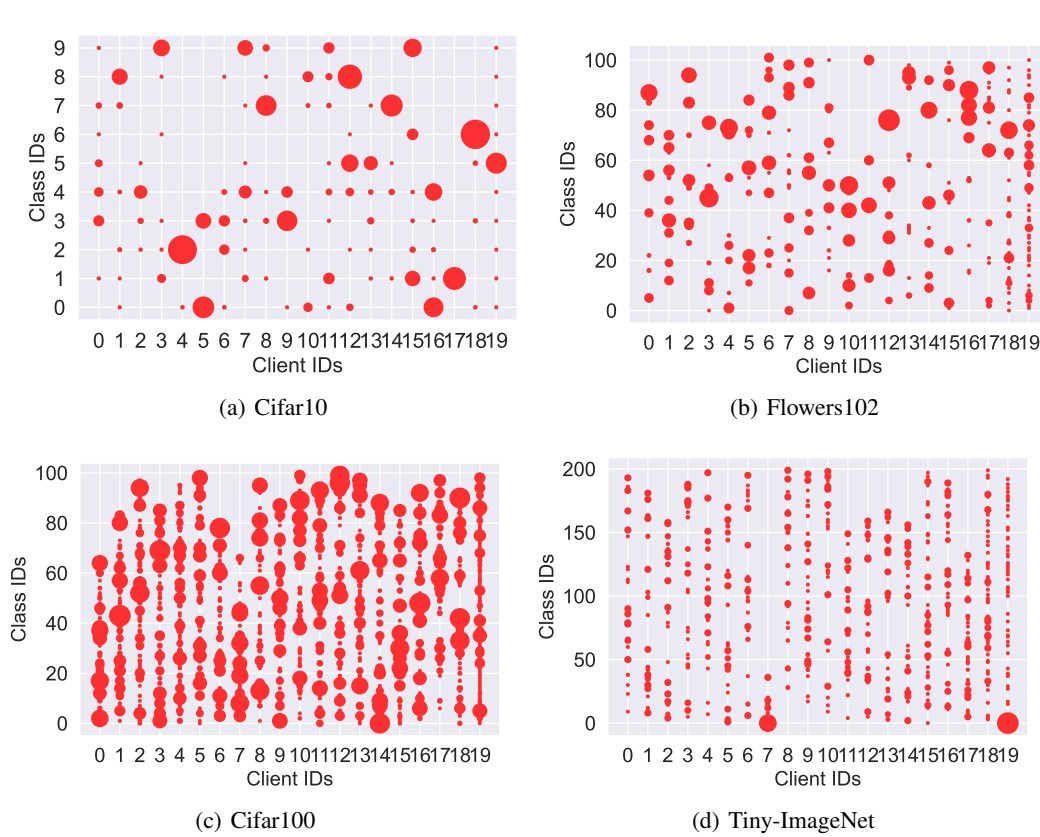

Figure 6: The data distribution of each client on Cifar10 ($\beta = 0.1$), Flowers102 ($\beta = 0.01$), Cifar100 ($\beta = 0.1$), and Tiny-ImageNet ($\beta = 0.01$), respectively, in Dirichlet setting s. The size of a circle represents the number of samples.

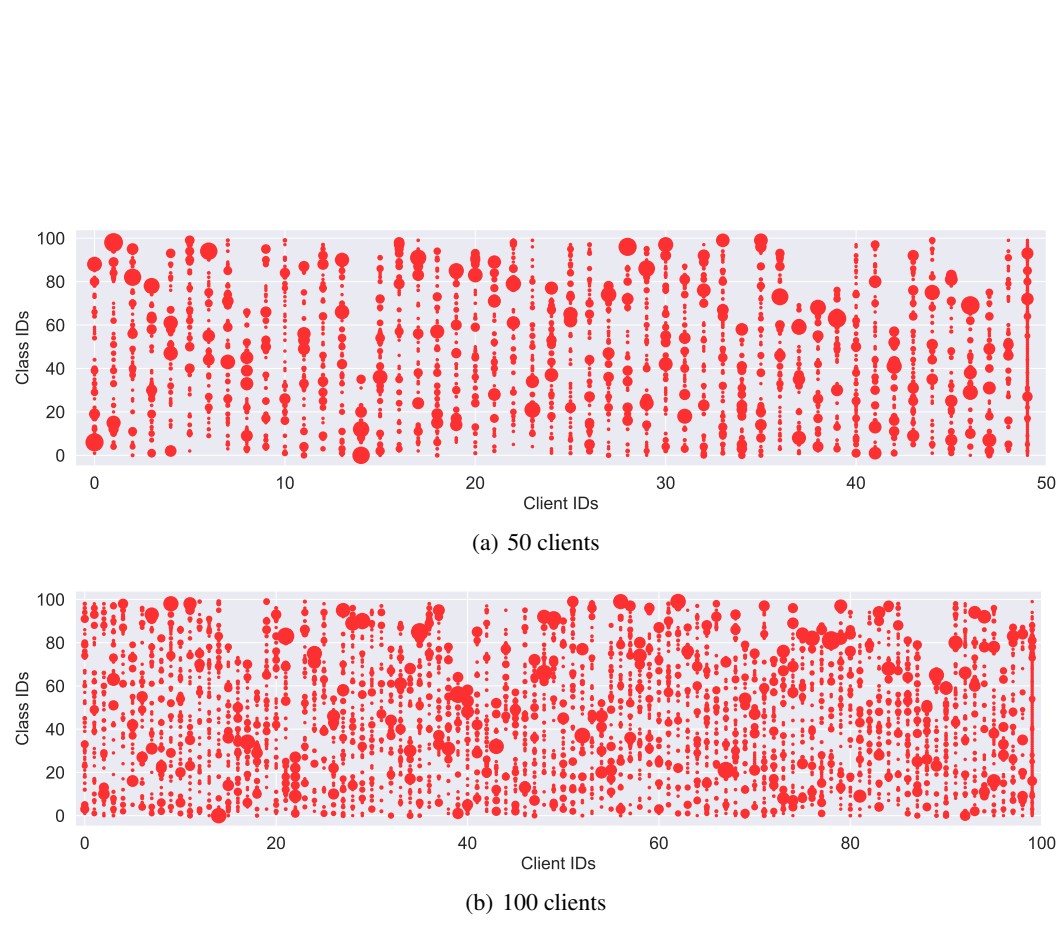

(a) 50 clients

(b) 100 clients

Figure 7: The data distribution of each client on Cifar100 in the Dirichlet setting ($\beta = 0.1$) with 50 and 100 clients, respectively. The size of a circle represents the number of samples.

