# OpenReview forum: "FedL2G: Learning to Guide Local Training in Heterogeneous Federated Learning"
_ICLR.cc/2025/Conference — Submitted to ICLR 2025_

### Official Review · Reviewer_UWko · 2024-10-20

**Soundness:** 3
**Presentation:** 3
**Contribution:** 2
**Rating:** 5
**Confidence:** 4

**Summary:**

This paper focuses on two classic challenges in Federated Learning (FL): data heterogeneity and model heterogeneity, both of which are crucial for deploying FL in real-world scenarios. In model-heterogeneous settings, aggregating model parameters is not feasible, making the aggregation of class prototypes a common approach. However, the authors observe that the simple aggregation of locally uploaded class prototypes into a global prototype fails to effectively guide the local training process. Thus, the authors propose a Federated Learning-to-Guide (FedL2G) method, which adaptively learns to guide local training in a federated manner and ensures that the additional guidance is beneficial to clients’ original tasks.

**Strengths:**

1) The paper tackles two critical challenges for deploying Federated Learning: data heterogeneity and model heterogeneity, which are essential for real-world FL deployment.
2) The authors identify that simple aggregation of class prototypes can even harm local updates, and therefore propose a learnable approach to generate global prototypes that effectively guide local training processes.
3) The paper provides a convergence proof, validating the theoretical feasibility of the proposed method.
4) Extensive experiments are conducted to demonstrate the effectiveness of FedL2G.

**Weaknesses:**

1) The meaning conveyed by Figure 1 is unclear. It lacks sufficient explanation, making it difficult to understand the challenges the paper aims to address based on the illustration.
2) Figure 2 has poor readability, and the lines are overly complicated and cluttered, making it hard to capture the key points. I recommend adding some descriptions to aid in understanding.
3) Although Figure 1 somewhat points out the challenges to be solved, the paper lacks sufficient motivation to validate this phenomenon.
4) I am concerned about whether uploading local class prototypes or features of local data samples may leak local data privacy since these data represent local statistical information.
5) There is a misuse of symbols. In Equation 1, the denominator uses "n" to represent the overall data, which may be misunderstood as "n" representing a specific device, especially since there are a total of N clients.
6) There is a lack of qualitative or quantitative analysis to prove that the proposed method effectively reduces the cross-entropy loss on users' local data.
7) Equation 6 aggregates gradients uploaded by all clients in a simple manner, which I believe may further introduce bias. This is because the information uploaded by strong clients (with stronger feature extraction capabilities and more data) should be more valuable than that from weaker clients.

**Questions:**

1) For the prototypes uploaded by different clients, can dynamic aggregation be performed based on the performance of each client? Strong clients often have richer information, so the initial global prototype could be aggregated based on their information to guide the training of weaker clients. As weaker clients produce more robust prototypes, they can also upload their local class prototypes to aggregate a new global prototype.
2) For other issues, see weakness.

---

> ### Author Response · Authors · 2024-11-23
> **Response (1/2)**
>
> We appreciate the reviewer's recognition of the value, theoretical analysis, and sufficient experiments of our work. Below, we reply to the identified weaknesses and questions. *Lines xxx* and *Section x* correspond to specific lines and sections in our paper.
>
>
> **Response to weakness 1: Figure 1 is unclear**.
>
> - We appreciate the reviewer's feedback regarding Figure 1. In the revised manuscript, we have updated Figure 1 and its corresponding main text to make the challenges addressed in this work more intuitive and easier to understand. Key elements have been clarified and emphasized.
> The revised text and figure are highlighted in blue for the reviewer's convenience.
>
>
> **Response to weakness 2: Figure 2 is complicated**.
>
> - We acknowledge the reviewer's concerns about the complexity of Figure 2. In the revised version, we have simplified Figure 2 by focusing on the key contributions (steps 3 and 4 of our FedL2G framework) and removing unnecessary lines. The figure caption has been expanded to include detailed explanations, ensuring readers can easily follow the processes depicted.
>
>
> **Response to weakness 3: the paper does not validate the phenomenon illustrated by Figure 1**.
>
> - As discussed in *Section 4.5* of the original manuscript, the phenomenon of objective mismatch is validated through experiments. Specifically, Figure 3 demonstrates how the original local loss (cross-entropy loss) fluctuates during federated training in methods like FedProto and FedDistill due to the objective mismatch problem, resulting in inferior model performance.
>
>
> **Response to weakness 4: uploading local class prototypes or features of local data samples may leak local data privacy**.
>
> - **Clarification**: Our FedL2G method does not upload raw features or local class prototypes. Instead, it uploads gradients of guiding vectors, as shown in *Line 12 of Algorithm 1*, which are initialized randomly and iteratively refined through client feedback. These gradients are not directly related to sensitive local data or class-specific statistical information.
> - **Privacy Analysis**: In *Section 4.6 FedL2G Protects Feature Information*, we provide a detailed discussion on this topic. Figure 4 demonstrates that guiding vectors differ significantly from class prototypes, ensuring privacy.
> - **Enhanced Privacy**: To further enhance privacy, we incorporated Gaussian noise into the gradients of guiding vectors following the approach in FedPCL [1]. FedL2G retains strong performance while improving privacy protection (see Table 1).
>
> Table 1: Test accuracy on Cifar100 in the Dirichlet setting using HtFE$_8$ with a noise scale of s and perturbation coefficient p.
>
> |                 | Original | Add Noise (s=0.05, p=0.1) | Add Noise (s=0.05, p=0.2) |
> |-----------------|:--------:|:---------:|:---------:|
> | Our FedL2G-l | 42.3     | 41.8      |  41.7      |
> | Our FedL2G-f | 43.8     | 42.9      | 42.6      |
>
>
> [1] Tan Y, Long G, Ma J, et al. Federated learning from pre-trained models: A contrastive learning approach. NeurIPS, 2022.
>
>
> **Response to weakness 5: misuse of symbol n**.
>
> - In the revised manuscript, we have replaced  "n"  with  "D" to clearly denote the total data size, distinguishing it from the number of clients  N .
>
>
> **Response to weakness 6: There is a lack of qualitative or quantitative analysis to prove that the proposed method effectively reduces the cross-entropy loss on users' local data**.
>
> - In *Section 4.5 FedL2G Prioritizes the Original Task*, we present both qualitative and quantitative analyses. Results show that FedL2G-l and FedL2G-f consistently reduce the original local cross-entropy loss, addressing the objective mismatch problem and enhancing local model performance.

---

> > ### Author Response · Authors · 2024-11-23
> > **Response (2/2)**
> >
> > **Response to weakness 7 and question 1: aggregating gradients uploaded by all clients may further introduce bias and strong clients often have richer information to facilitate weak clients**.
> >
> > - Without loss of generality, following common practice [1,2], our paper adopts a straightforward method to aggregate all uploaded gradients. Addressing inter-client differences is beyond the primary focus of our work.
> > - Directly using information from strong clients is not guaranteed to be suitable for weak clients. Addressing this issue would require additional efforts or even a new method [3], which falls beyond the scope of this paper. Weak clients are often better suited to assist one another [4], a principle central to the concept of clustered federated learning.
> > - We believe that exploring contribution-aware weighted aggregation would be an interesting avenue for future research. Nonetheless, **the primary contribution of this work lies in addressing the objective mismatch problem and demonstrating the efficacy of FedL2G, utilizing a simple yet widely adopted aggregation technique.**
> >
> >
> > [2] Ye M, Fang X, Du B, et al. Heterogeneous federated learning: State-of-the-art and research challenges[J]. ACM Computing Surveys, 2023, 56(3): 1-44.
> >
> > [3] Liu R, Wu F, Wu C, et al. No one left behind: Inclusive federated learning over heterogeneous devices. KDD, 2022.
> >
> > [4] Sattler F, Müller K R, Samek W. Clustered federated learning: Model-agnostic distributed multitask optimization under privacy constraints[J]. IEEE transactions on neural networks and learning systems, 2020, 32(8): 3710-3722.

---

### Official Review · Reviewer_dHdr · 2024-10-22

**Soundness:** 2
**Presentation:** 2
**Contribution:** 3
**Rating:** 5
**Confidence:** 4

**Summary:**

To address the objective mismatch in Heterogeneous Federated Learning, this paper proposes a Federated Learning-to-Guide method that adaptively learns to guide local training in a federated manner and ensures the extra guidance is beneficial to clients’ original tasks. The technique efficiently implements the learning-to-guide process using only first-order derivatives and achieves a non-convex convergence rate of $\mathcal{O}(1/T)$. The authors provide empirical validations on the theoretical results as well.

**Strengths:**

The proposed method is well-motivated, the paper shows that existing methods suffer from the objective mismatch issue, and show how to fix it.

The empirical results show that Federated Learning-to-Guide method is better than other lightweight HtFL methods, as expected, and the authors conduct ablations that show the influence of the server learning rate and visualizations of data distributions.

**Weaknesses:**

The analysis only covers FedOpt with SGD as the optimizer. Still, recent work [1] shows a different combination (Nesterov-accelerated SGD as the outer optimizer and AdamW as the inner optimizer) is much better for practical performance.

In the theorems that are presented, summarizing the main insights of these theorems may be needed since currently they are just written as long paragraphs.

In experiments, the least partial client participation ratio is set as 0.5. In more realistic settings, the participation ratio is lower with more clients.

[1].	Arthur Douillard, Qixuan Feng, Andrei A. Rusu, Rachita Chhaparia, Yani Donchev, Adhiguna Kuncoro, Marc'Aurelio Ranzato, Arthur Szlam, Jiajun Shen. DiLoCo: Distributed Low-Communication Training of Language Models. arXiv:2311.08105.

**Questions:**

See in weaknesses.

---

> ### Author Response · Authors · 2024-11-22
> **Response**
>
> We appreciate the reviewer's recognition of the well-motivated objective mismatch issue and sufficient experiments of our work. Below, we reply to the identified weaknesses. *Lines xxx* and *Section x* correspond to specific lines and sections in our paper.
>
> **Response to weakness 1: extend theoretical analysis on a newly proposed optimizer besides SGD**.
>
> - Our theoretical analysis focuses on the fundamental and widely-used SGD optimizer, aligning with most FL research. This approach ensures our analysis is general and adaptable, making it extensible to other specific optimizers. While the suggested combination of Nesterov-accelerated SGD (outer) and AdamW (inner) is effective for distributed low-communication training, it is tailored for a specialized use case rather than standard FL for distributed training of a large model. We believe maintaining the focus on SGD strengthens the generalizability of our contributions, but we are open to exploring other optimizer settings in future work.
>
>
> **Response to weakness 2: missing insights for theorems**.
>
> - **Key Insights**: The primary purpose of our theorems is to establish the non-convex convergence of FedL2G at a convergence rate of O(1/T), as highlighted in *Section Abstract*, *Section Introduction*, *Line 316*, and *Line 917*. This result demonstrates the theoretical soundness and efficiency of FedL2G in heterogeneous FL settings.
>
>
> **Response to weakness 3: client participation ratio**.
>
> - **Adherence to Existing Standards**: In our current experiments, we evaluate FedL2G using a client participation ratio (ρ) of 0.5, which is a widely accepted standard in FL research [1,2].
> - **Additional Results for Lower Participation Ratios**: As suggested by the reviewer, *"the participation ratio is lower with more clients,"* we conducted additional experiments with ρ=0.1 (10% client participation) for 100 clients. For comparison, our manuscript includes experiments with ρ=0.5 for 50 clients and ρ=1 for 20 clients. As shown in Table 1, FedL2G continues to outperform all baselines, demonstrating its robustness and scalability to lower participation ratios.
>
> Table 1: Test accuracy on Cifar100 in the Dirichlet setting using HtFE$_8$ with ρ=0.1.
>
> |  | 100 clients (ρ=0.1) |
> |--------------|:--------------:|
> |LG-FedAvg|41.0
> |FedGH|40.3
> |FML|35.2
> |FedKD|36.5
> |FedDistill|41.2
> |FedProto|28.6
> |Our FedL2G-l |41.6
> |Our FedL2G-f |**42.3**
>
> [1] Zhang J, Liu Y, Hua Y, et al. Fedtgp: Trainable global prototypes with adaptive-margin-enhanced contrastive learning for data and model heterogeneity in FL. AAAI, 2024.
>
> [2] Li Q, Diao Y, Chen Q, et al. Federated learning on non-iid data silos: An experimental study.ICDE, 2022.

---

### Official Review · Reviewer_xLkd · 2024-11-03

**Soundness:** 3
**Presentation:** 3
**Contribution:** 2
**Rating:** 6
**Confidence:** 4

**Summary:**

The paper under review studies heterogeneous federated learning (HtFL) and aims to address the mismatch between local and global learning objectives. By learning a set of local guiding vectors during model training, the local loss is prioritized when minimizing the global loss. The guiding vectors are compact in size and are updated based on feedback from clients’ local quiz sets. Therefore, the proposed FedL2G method is claimed to be lightweight, efficient, and adaptable.

**Strengths:**

1. Sections 1 and 2 clearly introduce the research problem, relevant literature, and contributions, making the paper easy to follow.

2. The mismatch between model personalization and generalization is identified as a primary research problem in the field of HtFL.

3. Guiding vectors, which are compact in size, are communicated between clients, thereby reducing communication overhead compared to direct model sharing.

4. Relevant benchmarks and theoretical analysis are included to support the paper’s contributions.

**Weaknesses:**

1. Feature Extraction Consistency: The consistency of feature extraction is not discussed in detail. With heterogeneous models and quiz sets across different clients, distributed feature extraction may produce varying representations (i.e., guiding vectors) for the same class. How can the averaging of vectors in line 13 of Algorithm 1 be effective in this context? Are there any explanations of feature extraction consistency when local models and private datasets are heterogeneous?

2. Warm-Up Period in Large-Scale Distributed Learning: In practical settings, a warm-up period may not be feasible in large-scale distributed learning scenarios. Does the warm-up phase require all clients to join the system for at least 50 rounds, as indicated on line 236? An ablation experiment should be added to help readers understand why the warm-up is necessary or to quantify the accuracy loss if it is omitted.

3. Data Heterogeneity Settings: The data heterogeneity settings may be overly strict. Using a Dirichlet distribution with control parameters like 0.1 and 0.01 makes the dataset highly skewed. The authors should explore a range of settings, from 0.01 to 1, to show how data heterogeneity impacts the performance of FedL2G.

4. Quiz Set Requirement: The proposed approach assumes that each client can maintain a quiz set, which implies that clients have sufficient training data. The authors should clarify a practical scenario to support this assumption and ensure that the settings are realistic.

**Questions:**

1. How can the averaging of vectors in line 13 of Algorithm 1 be ensured to be effective? Are there any explanations regarding feature extraction consistency when local models and private datasets are heterogeneous? The authors should provide quantitative analysis to assess how the consistency of feature extraction impacts the performance of the guiding vectors, for example, measuring cosine similarity between guiding vectors from different clients for the same classes, or examining how variance in feature representations correlates with model performance.

2. Does the warm-up phase require all clients to join the system for at least 50 rounds, as indicated on line 236? An ablation experiment should be included to help readers understand the necessity of the warm-up phase or to quantify the accuracy loss if it is omitted. The review suggests the authors compare performance with different warm-up durations (e.g., 0, 25, 50, 100 rounds) and analyze how it affects convergence speed and final accuracy across various client participation scenarios.

3. Will the proposed method outperform benchmarks in both homogeneous and heterogeneous data settings? The authors should explore a range of heterogeneity settings, with a verity of Dirichlet parameters, to demonstrate how data heterogeneity impacts the performance of FedL2G. The reviewer suggests adopting specific Dirichlet parameter values to test (e.g., α = 0.01, 0.1, 0.5, 1, 10) and suggests key metrics to report for each setting, such as accuracy, convergence speed, and communication efficiency.

4. What real-world scenario allows for sufficient quiz sets and warm-up phases? The authors should clearly state a practical scenario to justify the feasibility of these settings. The reviewer suggests that the authors may provide specific examples of real-world applications where these requirements could be met, or discuss potential modifications to the method for scenarios with limited data or stricter time constraints.

**Details Of Ethics Concerns:**

No ethics concerns are raised for this paper.

---

> ### Author Response · Authors · 2024-11-22
> **Response (1/2)**
>
> We appreciate the reviewer's recognition of the clarity, value, efficiency of meta-learned guiding vectors, and sufficient emprical and theoretical analysis of our work. Below, we reply to the identified weaknesses and questions. *Lines xxx* and *Section x* correspond to specific lines and sections in our paper.
>
> **Response to weakness 1 and question 1: Feature Extraction Consistency**.
>
> - **Effectiveness of Averaging**: Averaging, as introduced in FedProto [1], is a widely accepted and effective practice in FL for aggregating and sharing global information under both data and model heterogeneity. In our FedL2G framework, updating local models using global guiding vectors plays a crucial role in aligning local models and promoting consistency in their feature extraction. Without the global guiding vectors, local models lack this critical alignment, resulting in significantly poorer performance, as demonstrated in Table 1.
> - **Consistency Improvement During the FL Process**: As discussed above, aligning local models with global guiding vectors gradually enhances feature extraction consistency across heterogeneous models. This improvement is evidenced by the increased cross-client cosine similarity shown in Table 2.
>
> Table 1: Test accuracy on two datasets in the Dirichlet setting using HtFE$_8$.
>
> |  | Cifar10     | Cifar100    |
> |--------------|:--------------:|:--------------:|
> |Locally training without global guiding vectors |83.2|35.6|
> |Our FedL2G-l |86.5|42.3|
> |Our FedL2G-f |**87.6**|**43.8**|
>
> Table 2: Cross-client cosine similarity (mean) of extracted features for the same class in FedL2G-f on Cifar10 under the Dirichlet setting with HtFE$_8$.
>
> |  | Round=0     | Round=10    | Round=20    | Round=50    |
> |--------------|:--------------:|:--------------:|:--------------:|:--------------:|
> |Cosine |0.5|0.7|0.8|0.9|
> |Accuracy |8.9|80.9|85.2|86.7
>
>
> [1] Tan Y, Long G, Liu L, et al. Fedproto: Federated prototype learning across heterogeneous clients. AAAI, 2022.
>
>
> **Response to weakness 2 and question 2: Ablation Study of Warm-Up Period**.
>
> - **Computational Efficiency**: The warm-up phase, which includes steps 1, 3, 4, 5, 6, and 7, is computationally lightweight and closely mirrors the main FL process, with the exception that step 2 (local model updates) is skipped (see *Lines 233-240*). This design ensures that the warm-up phase requires minimal additional effort.
> - **Scalability**: As shown in *Lines 221-225*, we **only require participating clients to join instead of all clients** in the warm-up phase.
> - **Ablation Study Results**: As shown in Table 3, FedL2G maintains competitive performance even with no warm-up (T'=0). **The introduction of the warm-up phase does not impact the overall convergence speed.** However, a short warm-up phase enhances guiding vector initialization, improving subsequent rounds' performance. Notably, FedL2G-f benefits more from a warm-up phase due to the higher learning capacity of the high-dimensional feature space. Overly large T' negatively impacts both variants due to overfitting on untrained client models.
>
> Table 3: Ablation study of the number of warm-up rounds (T') on Cifar100 in the default Dirichlet setting using HtFE$_8$ with different T'. Results (a, b) represent (accuracy, the total number of converged rounds including the warm-up round). The total converged round represents convergence speed.
>
> |  | T'=0 (no warming-up)     | T'=1    | T'=10    | T'=20    | T'=50    | T'=100    |
> |--------------|:--------------:|:--------------:|:--------------:|:--------------:|:--------------:|:--------------:|
> |Our FedL2G-l |41.7 (160)|41.8 (156)|41.7 (165)|42.0 (158)|42.3 (159)|41.8 (161)|
> |Our FedL2G-f |40.9 (163)|41.6 (160)|43.0 (155)|43.6 (157)|43.8 (160)|43.6 (162)|

---

> > ### Author Response · Authors · 2024-11-22
> > **Response (2/2)**
> >
> > **Response to weakness 3 and question 3: Various Data Heterogeneity Settings**.
> >
> > - **Existing Results**: In *Section 4*, we evaluated FedL2G under three levels of data heterogeneity: pathological, Dirichlet (β=0.1), and Dirichlet (β=0.01). These are standard settings for studying data heterogeneity [2].
> >
> > - **Additional Experiments**: As suggested by the reviewer, we conducted experiments using β values of 0.01, 0.5, and 1. Table 4 demonstrates that FedL2G consistently outperforms baselines across all settings, even as data heterogeneity varies. While larger β results in less skewed data distributions, it reduces per-class data availability for clients, impacting overall performance. The communication efficiency remains consistent across different scenarios, as the gradients of the guiding vectors retain the same shape in every communication round.
> >
> > Table 4: Test accuracy on Cifar100 in the Dirichlet setting using HtFE$_8$ with varying β. The results in "()" means the total number of converged round including the warm-up phase for FedL2G.
> >
> > |  | β=0.01 | β=0.1 | β=0.5 | β=1 |
> > |---|:---:|:---:|:---:|:---:|
> > |LG-FedAvg|66.6 (178)|40.7 (190)|21.3 (273)|15.7 (141)|
> > |FedGH|65.2 (146)|41.0 (226)|21.2 (232)|15.5 (184)|
> > |FML|64.5 (370)|39.9 (287)|20.0 (150)|16.0 (318)|
> > |FedKD|64.9 (285)|40.6 (198)|21.5 (166)|16.3 (288)
> > |FedDistill|67.0 (338)|41.5 (216)|22.1 (161)|16.4 (273)
> > |FedProto|60.6 (540)|36.3 (533)|18.3 (570)|12.6 (369)
> > |Our FedL2G-l |**68.2 (196)**|**42.3 (176)**|**22.1 (189)**|**16.7 (172)**
> > |Our FedL2G-f |**70.6 (257)**|**43.8 (235)**|**23.3 (225)**|**16.8 (210)**
> >
> >
> > [2] Zhang J, Hua Y, Wang H, et al. Fedala: Adaptive local aggregation for personalized federated learning. AAAI 2023.
> >
> >
> > **Response to weakness 4 and question 4: Quiz Set Requirement**.
> >
> > - **Practical Feasibility**: The quiz set is a small portion held out from the original training data (see *Lines 65-67*). In real-world scenarios, most clients possess sufficient data to allocate a small batch (batch size = 10) as a quiz set. Clients with extremely limited data are unlikely to participate in FL effectively [3]. Given that a minimal quiz set is sufficient, our approach is as practical as a conventional FL approach.
> >
> > - **Robustness**: As shown in Table 5, even with a reduced quiz set size of 2 or 5 samples, FedL2G achieves strong performance, demonstrating its robustness to small quiz sets.
> >
> > Table 5: Test accuracy on Cifar100 in the Dirichlet setting using HtFE$_8$ with different quiz set size (qss).
> >
> > |  | original (qss=10) | qss=2  | qss=5|
> > |--------------|:--------------:|:--------------:|:--------------:|
> > |Our FedL2G-l |42.3|42.2|42.3|
> > |Our FedL2G-f |43.8|44.2|43.4|
> >
> > [3] Gouissem A, Chkirbene Z, Hamila R. A comprehensive survey on client selections in federated learning[J]. Innovation and Technological Advances for Sustainability, 2024: 417-428.

---

### Official Review · Reviewer_UTFs · 2024-11-04

**Soundness:** 2
**Presentation:** 2
**Contribution:** 2
**Rating:** 6
**Confidence:** 3

**Summary:**

The paper introduces the FedL2G method, designed to address challenges in Heterogeneous Federated Learning that arise from data and model heterogeneity. The proposed method focuses on optimizing the training process by ensuring that additional guiding objectives introduced during the federated learning process are beneficial and align with clients' original local objectives.

**Strengths:**

1. The paper is well-written and easy to follow.
2. The paper is well-motivated that Heterogeneous Federated Learning is quite realistic.
3. The authors provide rigorous convergence analysis.

**Weaknesses:**

1. The paper may overlook some important related methodologies, such as FedPCL[1] and FPL[2], which also address the central issue highlighted by the authors: the deviation of aggregated global prototypes from client-specific feature vectors due to data heterogeneity. Specifically, FPL tackles similar challenges, making its comparison with FedL2G pertinent for a comprehensive evaluation.
2. The proposed method introduces complexity by requiring an additional small quiz set and a warm-up period, which could complicate implementation. The quiz set, while beneficial for validating model updates, may also introduce an unfair advantage over other baselines that do not use this approach.
3. It would be beneficial for the authors to include more recent federated learning methodologies like FedPCL[1] and FPL[2] in the experimental comparisons. The current baseline methods, while foundational, may not represent the state-of-the-art, limiting the robustness of the comparative analysis presented.
4. Including experiments on the DominNet dataset could enhance the applicability and robustness of the FedL2G method across diverse scenarios.

[1] Tan, Y., Long, G., Ma, J., Liu, L., Zhou, T., Jiang, J.: Federated learning from pre-trained models: A contrastive learning approach. Advances in Neural Information Processing Systems 35, 19332–19344 (2022)

[2] Huang, W., Ye, M., Shi, Z., Li, H., Du, B.: Rethinking federated learning with domain shift: A prototype view. In: 2023 IEEE/CVF Conference on Computer Vision and Pattern Recognition (CVPR). pp. 16312–16322. IEEE (2023)

**Questions:**

Please see the weakness above.

---

> ### Author Response · Authors · 2024-11-22
> **Response (1/2)**
>
> We appreciate the reviewer's recognition of the clarity, value, and rigorous convergence analysis of our work. Below, we reply to the identified weaknesses. *Lines xxx* and *Section x* correspond to specific lines and sections in our paper.
>
> **Response to weakness 1: Comparison to FedPCL and FPL**.
>
> FedPCL and FPL differ from our FedL2G in both scenario and objective, and their direct application is inapplicable in our model heterogeneity setting. However, we have modified these methods to enable a fair comparison and included results in Table 1.
>
> - **Scenario**: Both FedPCL and FPL focus solely on data heterogeneity with **homogeneous client model architectures**. Specifically,
>   - FedPCL requires pretrained and frozen backbones shared among clients and only trains homogeneous projection layers (two fully-connected layers) locally.
>   - FPL is tailored for domain-shift settings and aggregates homogeneous client models into a global model during each communication round.
>   - In contrast, FedL2G is designed for **practical scenarios with both data and model heterogeneity**, enabling collaboration among diverse architectures such as shallow CNNs, GoogleNet, MobileNet, ResNets, and ViT.
> - **Objective**:
>   - Both FedPCL and FPL focus on adjusting local features according to the global/cluster prototypes (aggregated/existing feature information extracted by client models), where local features can be **misled by poor prototypes** in early FL rounds.
>   - FedL2G instead learns **guiding information** (independent of extracted features) to **meta-learn guidance that minimizes local loss during each FL round**. This prioritization of local objectives differentiates FedL2G from FPL's prototype-based approach.
> - **Comparison**: We modified FedPCL and FPL for use in our scenario:
>   - FedPCL*: Trains the entire local model from scratch.
>   - FPL*: Omits global model aggregation due to model heterogeneity.
>   Table 1 shows their relatively poor performance, reflecting their limitations in model heterogeneity settings and the amplification of poor prototypes by their contrastive learning strategies.
>
> Table 1: Test accuracy on four datasets in the Dirichlet setting using HtFE$_8$.
>
> |  | Cifar10     | Cifar100    | Flowers102    | Tiny-ImageNet |
> |--------------|:--------------:|:--------------:|:--------------:|:--------------:|
> |FedPCL* |70.5|27.3|51.7|21.9|
> |FPL* |81.7|34.2|60.4|37.7|
> |FedMRL (NeurIPS 2024) |86.2|41.2|70.1|48.2|
> |Our FedL2G-l |86.5|42.3|71.5|49.5|
> |Our FedL2G-f |**87.6**|**43.8**|**73.6**|**50.3**|
>
>
> **Response to weakness 2: Using small quiz set and warm-up period is complex**.
>
> - **Quiz set**: The quiz set is not an additional dataset but a small portion held out from the original training data (see *Lines 65-67*), ensuring fairness and no extra data advantage over other baselines. As shown in Table 2, only 2 to 5 samples are sufficient for our FedL2G to achieve strong performance. This implementation is straightforward and supported by tools like [higher](https://github.com/facebookresearch/higher).
> - **Warm-up period**: The warm-up phase (containing step 1, 3, 4, 5, 6, 7) is computationally lightweight and identical to the main FL process, except for skipping step 2 local model updates (see *Lines 233-240*). It requires minimal additional effort. Additionally, our FedL2G-l achieves an accuracy of 41.7 with (T'=0) (i.e., no warm-up), and FedL2G-f achieves 41.6 with (T'=1).
>
> Table 2: Test accuracy on Cifar100 in the Dirichlet setting using HtFE$_8$ with different quiz set size (qss).
>
> |  | original (qss=10) | qss=2  | qss=5|
> |--------------|:--------------:|:--------------:|:--------------:|
> |Our FedL2G-l |42.3|42.2|42.3|
> |Our FedL2G-f |43.8|44.2|43.4|
>
>
>
> **Response to weakness 3: Adding more recent relevant FL methodologies**.
>
> - At submission, the most recent baseline identified was FedGH (2023), which we compared extensively (*Section 2.1 & 4*). These comparisons already demonstrate FedL2G's strong performance in handling data and model heterogeneity.
> - We have now incorporated FedPCL and FPL comparisons (as detailed above). Additionally, we include results for FedMRL [1], a newly identified baseline (NeurIPS 2024). Our FedL2G maintains its superiority in Table 1, even when FedMRL utilizes both the local model and an auxiliary global model for inference.
>
> [1] Yi L, Yu H, Ren C, et al. Federated Model Heterogeneous Matryoshka Representation Learning. NeurIPS, 2024.

---

> > ### Author Response · Authors · 2024-11-22
> > **Response (2/2)**
> >
> > **Response to weakness 4: Including experimental results on DomainNet**.
> >
> > - While experiments on Cifar10, Cifar100, Flowers102, and Tiny-ImageNet already establish the generality of FedL2G, we appreciate the reviewer's suggestion to test on DomainNet. We have now included these results to further validate our method's applicability and robustness (Table 3).
> >
> > Table 3: Test accuracy on DomainNet using HtFE$_4$.
> >
> > |  | DomainNet |
> > |--------------|:--------------:|
> > |LG-FedAvg|26.9|
> > |FedGH|25.0|
> > |FML|24.9|
> > |FedKD|25.0|
> > |FedDistill|26.8|
> > |FedProto|21.2|
> > |Our FedL2G-l |27.3|
> > |Our FedL2G-f |**28.2**|

---

### Author Response · Authors · 2024-11-26
**Thank you for your valuable feedback and insights**

Dear Reviewers,

Thank you for your dedicated time and effort in reviewing our work. As the discussion phase nears its end, we kindly invite your feedback on our responses to ensure we’ve thoroughly addressed your concerns.

If you have any additional questions or points for discussion, we’d be happy to engage further. Your input is invaluable, and we deeply appreciate your consideration.

Best regards,
The Authors

---

### Author Response · Authors · 2024-11-30
**Sincere Thanks for Reviewers' Constructive Feedback**

Dear Reviewers,

We would like to express our heartfelt gratitude for the time and effort you have dedicated to reviewing our paper. Your thoughtful comments (**mainly focused on adding auxiliary experiments for further improvement**) have been invaluable in improving both the clarity and quality of our work. For your convenience, we’ve summarized our rebuttal below.

### **Contributions of our Paper**

1. We observed a negative effect—an increase in the **original local loss**—resulting from the use of an additional **guiding objective** during each client’s local training in representative prototype-based methods in HtFL, particularly under data and model heterogeneity. We analyzed this phenomenon and attributed it to a **mismatch** between the prototype-guiding objective and the client's original local objective.
2. To address the objective mismatch issue, we propose a novel HtFL method, FedL2G, which **prioritizes the original local objective while incorporating the guiding objective**. Instead of using the extracted and aggregated feature vectors (i.e., prototypes) in the guiding objective, we learn the guiding vectors from scratch using a meta-learning approach. In other words, **the semantics of prototypes and guiding vectors differ**. While prototypes are feature vectors, **guiding vectors are not**—they may lie outside the feature space of the client models. **The semantics of guiding vectors can be understood as "learned vectors designed to further reduce the original local loss."** This is **a common expectation among all clients, making it worthwhile to share.**
3. We provide **theoretical guarantees** and efficiently implement the learning-to-guide process using only first-order derivatives with respect to model parameters. This approach achieves a **non-convex convergence rate of O(1/T)**.
4. We conduct experiments across **two data heterogeneity** and **six model heterogeneity** settings, enabling collaborative learning across **14 heterogeneous model architectures (e.g., ResNets and ViTs)** within a single FL system. Our method demonstrates superior performance compared to six heterogeneous-model-applicable state-of-the-art approaches.


### **Summary of our rebuttal**

1. **Experiments with related approaches.** We included a comprehensive set of widely used heterogeneous-model-applicable state-of-the-art baselines in our experiments at the time of submission. During the rebuttal, we also compared our method with the suggested homogeneous-model approaches, FedPCL and FPL, as well as a newly accepted method, FedMRL.
2. **Experiments on more datasets and settings.** During the rebuttal, we included additional experiments on DomainNet, exploring different settings for the warm-up round (T'), varying degrees of data heterogeneity (β), different quiz set sizes (qss), and varying client participation ratios (ρ) across 100 clients. We also introduced noise to further enhance privacy, with negligible impact on performance.
3. **Further clarification on the quiz set and warm-up period.** The **quiz set is not an additional dataset** but a small portion held out from the original training data (see Lines 65-67), ensuring fairness without providing any extra data advantage over other baselines. **Only 2 to 5 samples are sufficient** for our FedL2G to achieve strong performance. The warm-up phase (steps 1, 3, 4, 5, 6, 7) is computationally lightweight and mirrors the main FL process, except for skipping step 2 (local model updates) (see Lines 233-240), requiring minimal additional effort. Furthermore, our FedL2G-l achieves an accuracy of **41.7 with (T' = 0) (i.e., no warm-up)**, while FedL2G-f achieves **41.6 with (T' = 1)** on Cifar100 in the default Dirichlet setting using HtFE$_8$.
4. **Improve the readability of Figures 1 and 2, as well as the notations.** We thank the reviewers for their constructive feedback on readability, and we have revised the paper accordingly in the newly uploaded version.

If you have any further concerns, we would be more than happy to address them :-).

Best regards,

The Authors

---

### Meta-Review · Area_Chair_9ZtJ · 2024-12-08

**Metareview:**

The paper introduces the FedL2G method, designed to address challenges in Heterogeneous Federated Learning that arise from data and model heterogeneity. The proposed method has theoretical guarantees, and have been shown to outperform prior works.

While the approach is supported by theoretical guarantees, the strength of these results is limited. This is because a non-collaborative setting could achieve similar guarantees. It is recommended to include such a baseline and provide a comparative discussion in the revision. A promising direction for future work would be to demonstrate convergence speedup as the number of agents increases, akin to the linear speedup observed with homogeneous agents. Additionally, it may be valuable to introduce a parameter to quantify agent heterogeneity, showing that as agents become more homogeneous, the convergence rate improves.

**Additional Comments On Reviewer Discussion:**

NA

---

> ### Public Comment · ~Jianqing_Zhang1 · 2025-02-10
> **Reply to Meta Review by Area Chair 9ZtJ**
>
> We sincerely appreciate your time and effort in reviewing our paper and recognizing its contributions. However, we would like to clarify two misunderstandings in your comments:
>
> 1. **Collaborative Setting**:
>    Our theoretical analysis is indeed conducted within a *collaborative* setting, as detailed in **Appendix C**. Specifically, we follow the framework established in FedProto [1], proving convergence with a focus on individual client losses that include a collaborative term, $\mathcal{G}$, as shown in Equations (C.13) and (C.14). In the subsequent proofs, $\mathcal{G}$ is further expressed using the collaborative notation $\mathbb{E}_{i \sim [N]} \pi_i$. For simplicity, we omit the client ID $i$ in Equations (C.15–C.20), as mentioned in our manuscript. Therefore, our theoretical guarantees are firmly rooted in a collaborative setting.
>
> 2. **Agents vs. Federated Learning**:
>    Our work is centered on the federated learning domain, not on multi-agent systems. The focus is on client-server interactions typical of federated learning frameworks, rather than agent-based paradigms.
>
> We hope this clarification addresses your concerns and helps in further understanding our contributions. Thank you again for your thoughtful review.
>
> **Reference:**
> [1] Tan, Yue, et al. "FedProto: Federated Prototype Learning Across Heterogeneous Clients." *AAAI*, 2022.

---

### Decision · Program_Chairs · 2025-01-22

Reject